# PipeSD: An Efficient Cloud-Edge Collaborative Pipeline Inference Framework with Speculative Decoding

**Yunhe Han** [* 1]   **Yunqi Gao** [* 1 2]   **Bing Hu** [1]   **Mahdi Boloursaz Mashhadi** [3]   **Yitong Duan** [4]   **Pei Xiao** [3]
**Yanfeng Zhang** [2]

## Abstract

Speculative decoding can significantly acceler-
ate LLM inference, especially given that its
cloud-edge collaborative deployment offers cloud
workload offloading, offline robustness, and pri-
vacy enhancement. However, existing collab-
orative inference frameworks with speculative
decoding are constrained by (i) sequential to-
ken generation and communication with low re-
source utilization, and (ii) inflexible cloud non-
autoregressive verification (NAV) triggering that
induces premature verification or costly rollbacks.
In this paper, we propose *PipeSD*, an efficient
cloud-edge collaborative pipeline inference frame-
work with speculative decoding. PipeSD over-
laps token generation and communication by a
token-batch pipeline scheduling mechanism opti-
mized by dynamic programming, and improves
verification flexibility through a dual-threshold
NAV triggering mechanism with a lightweight
Bayesian optimization autotuner. We implement
PipeSD using llama-cpp-python, PyTorch, and
FastAPI, and evaluate it on a real-world cloud-
edge testbed with two draft-target model pairs
across four scenarios. Results show that PipeSD
consistently outperforms state-of-the-art base-
lines, achieving $1.16\times$–$2.16\times$ speedup and reduc-
ing energy consumption by 14.3%–25.3%. Our
code is available at https://github.com/
Ghanyunhe/PipeSD.

## 1. Introduction

Generative Large Language Models (LLMs) have achieved
remarkable success in tasks like dialogue, content creation,
and coding (Holmes et al., 2024; Grattafiori et al., 2024;
OpenAI et al., 2024). They are predominantly built on a
decoder-only transformer architecture, which employs au-
toregressive generation to produce output sequences, i.e.,
generating one token at a time (Brown et al., 2020; Touvron
et al., 2023a; Dao et al., 2022; Vaswani et al., 2017). How-
ever, the inherent sequential dependency of autoregressive
generation makes the inference process a significant latency
bottleneck. Speculative decoding has emerged as an efficient
approach to accelerate LLM inference without degrading
output quality (Leviathan et al., 2023; Chen et al., 2023).
The core idea is to employ a smaller, faster "draft" model
to speculate multiple draft tokens, which are subsequently
verified by a larger "target" model in a single forward pass,
thereby reducing inference latency while exactly preserving
the output distribution of the target model.

Existing studies have deployed speculative decoding in the
cloud (Cai et al., 2024; Li et al., 2025b; Zhao et al., 2024) or
on edge devices (Xu et al., 2025), both of which exhibit in-
herent limitations. Cloud-based deployment leverages pow-
erful hardware and diverse optimization frameworks (Kwon
et al., 2023; NVIDIA, 2025; Zheng et al., 2024), and en-
ables high-speed execution of large-scale models. However,
it depends on continuous network connectivity and raises
data privacy concerns (Zhan et al., 2025). In addition, edge
deployment supports offline inference and enhanced data
privacy (Xu et al., 2025), but is constrained by limited com-
putational resources and memory, hindering the efficient
deployment of LLMs.

Compared to the two deployment modes mentioned above,
the "draft-and-verify" architecture of speculative decoding
is inherently more suitable for cloud-edge collaborative in-
ference mode, which leverages the computing power of
both cloud and edge, along with communication between
them, to complete inference tasks (Li et al., 2025a; Tian
et al., 2024). Specifically, the draft model is deployed at the
edge for rapid autoregressive generation, while the target
model resides in the cloud for non-autoregressive verifica-

---
*Equal contribution [1]School of Information Science and Elec-
tronic Engineering, Zhejiang University, Hangzhou, China [2]School
of Computer Science and Engineering, Northeastern University,
Shenyang, China [3]5GIC & 6GIC, Institute for Communication Sys-
tems (ICS), University of Surrey, Guildford, UK [4]Zhongguancun
Institute of Artificial Intelligence, Beijing, China. Correspondence
to: Bing Hu <binghu@zju.edu.cn>.

*Proceedings of the $43^{rd}$ International Conference on Machine
Learning*, Seoul, South Korea. PMLR 306, 2026. Copyright 2026
by the author(s).

tion (NAV). This mode not only leverages edge resources to reduce cloud workloads but also provides an adaptive execution strategy, e.g., under unstable network conditions or for privacy-sensitive tasks, the edge device can autonomously switch to local inference. Some pioneering work has started exploring the potential of cloud-edge collaborative inference frameworks with speculative decoding, such as HSL (Hao et al., 2024), HAT (Xie et al., 2025), and SpecEdge (Park et al., 2025).

However, existing frameworks face two main limitations: (1) *Sequential token generation and communication.* Traditional frameworks execute token generation, communication, and cloud NAV sequentially, forcing the cloud to wait for the edge to generate and upload all draft tokens and forcing the edge to wait for NAV feedback, thereby underutilizing bandwidth and computing power. (2) *Inflexible NAV triggering mechanism.* Existing frameworks either adopt a fixed draft length for NAV, ignoring task complexity, or rely on single confidence-based triggering conditions, which can delay error detection or lead to excessive speculation.

To address the above limitations, we propose *PipeSD*, an efficient cloud-edge collaborative pipeline inference framework with speculative decoding. We make the following main technical contributions: (1) PipeSD introduces a token-batch pipeline scheduling mechanism that overlaps draft generation and communication to maximize resource utilization and minimize inference latency, which mathematically formulates pipeline scheduling as an optimization problem and obtains an optimal solution by dynamic programming (DP). (2) PipeSD adopts a dual-threshold NAV triggering mechanism to improve verification flexibility, by considering the confidence of both the overall sequence and individual tokens (token confidence is the probability assigned by the draft model to that token and sequence confidence is the product of the confidences of all tokens). It integrates a lightweight Bayesian optimization (BO) autotuner for automatic threshold adaptation. (3) We implement PipeSD using llama-cpp-python (Betlen, Andrei, 2023), PyTorch (Paszke et al., 2019), and FastAPI (FastAPI Developers, 2018), and evaluate it on real-world cloud-edge testbeds. Experiments on two draft-target model pairs across four scenarios demonstrate that PipeSD consistently outperforms state-of-the-art baselines, including HSL and EdgeLLM, achieving $1.16\times$–$2.16\times$ speedup and reducing energy consumption by 14.3%–25.3%.

## 2. Background and Challenges

### 2.1. Decoder-based LLMs and Autoregressive Inference

Generative LLMs predominantly adopt the decoder-only transformer architecture (Brown et al., 2020; Touvron et al., 2023a; Dao et al., 2022). This design employs a stack of

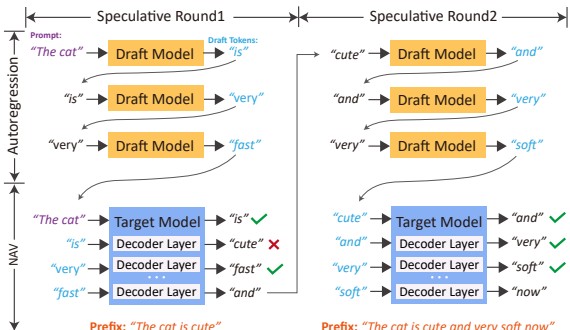

*Figure 1.* Illustration of the speculative decoding process.

decoder layers, each utilizing masked self-attention and feed-forward networks to process input sequences (Sheng et al., 2023). Decoder-only LLMs typically adopt autoregressive generation, which produces output sequence one token at a time, and each newly generated token is appended to the input sequence to predict the subsequent one (Vaswani et al., 2017; Radford et al., 2019). This inherent sequential dependency, however, makes the inference process a significant latency bottleneck.

### 2.2. Accelerating Inference with Speculative Decoding

Figure 1 illustrates an example of speculative decoding (Leviathan et al., 2023; Chen et al., 2023), which accelerates the inference process by using a draft model to predict a draft sequence (that consists of multiple draft tokens), and validating it in a single forward pass by the target model. We define a *speculative round* including two steps: First, the draft model autoregressively generates a sequence of draft tokens. Second, all draft tokens are sent to the target model for a single NAV. If the draft tokens match those the target model would generate, they are accepted. Otherwise, the target model corrects the first mismatched token, and the draft tokens before that token are accepted. The accepted and corrected tokens are appended to the output sequence and serve as the prefix for subsequent inference. During an inference task, speculative rounds repeat until the complete output accepted token sequence is produced. This method leverages the high probability that many tokens can be correctly predicted by a draft model on common tasks, thus breaking the strict autoregressive bottleneck of target model and significantly reducing end-to-end inference latency while preserving the output quality.

### 2.3. Cloud-Edge Collaborative Inference with Speculative Decoding

Speculative decoding is naturally well-suited for a cloud-edge collaborative inference architecture (Hao et al., 2024; Xie et al., 2025; Park et al., 2025), where the lightweight draft model is deployed on a resource-constrained edge de-

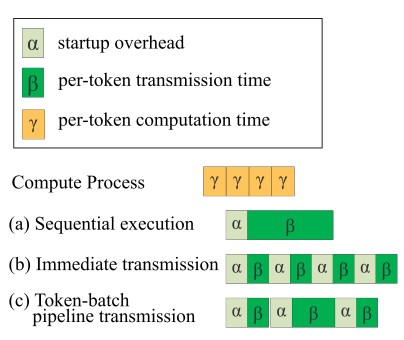

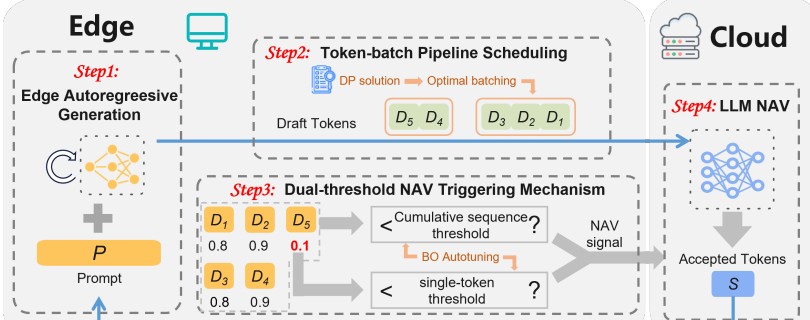

Figure 2. Comparison of transmission strategies.

Figure 3. Workflow of PipeSD within one speculative round.

vice (e.g., a smartphone), and the large target model resides in the cloud. In a speculative round of traditional collaborative inference with speculative decoding, the edge device first generates draft tokens autoregressively. Then, the edge transmits them to the cloud for NAV. Finally, the cloud validates the draft tokens using the target model and returns the accepted tokens. This distributed design offers three key advantages: (i) it effectively leverages the computational capability of edge devices to reduce the workload of the cloud; (ii) it enables the local draft model to continue executing lightweight tasks under poor network connectivity; and (iii) edge devices selectively upload inference tasks to the cloud, thereby improving data privacy.

### 2.4. Performance Bottlenecks

While cloud-edge collaborative inference with speculative decoding holds great promise, existing frameworks are constrained by two factors: (1) **Sequential computation-communication execution pattern.** Existing frameworks typically adopt a compute-first, transmit-later pattern (Hao et al., 2024). Specifically, the edge device generates the entire draft sequence and then transmits it to the cloud as shown in Figure 2(a). This sequential execution results in unnecessary idle time on both the edge and cloud sides, leading to low utilization of bandwidth and computing power; (2) **Inflexible NAV triggering mechanism.** Traditional speculative decoding generates a fixed number of draft tokens in each speculative round, ignoring the varying difficulty of inference tasks (Kim et al., 2023). This tends to result in either premature verification that misses speculation opportunities, or over-speculation that causes large-scale rollbacks. HSL introduces a single-token confidence-based NAV triggering mechanism (Hao et al., 2024), where NAV is triggered if the confidence of any draft token falls below a predefined threshold. However, this mechanism may never activate verification if each token appears moderately confident, causing over-generation. Moreover, EdgeLLM introduces a cumulative sequence confidence metric (Xu et al., 2025), which controls NAV triggering by monitoring the sequence confidence. Since the scheme accumulates the confidence of the

entire draft sequence, it may mask tokens with excessively low confidence, leading to delayed error detection. Beyond cloud-edge-specific frameworks, the broader speculative decoding literature has also explored more adaptive NAV triggering strategies. (Zhang et al., 2025) uses the entropy of the draft-tokens as an uncertainty signal to determine when to trigger NAV. (Huang et al., 2025) triggers NAV based on a learned prediction signal. Nevertheless, these methods still rely on a single signal to control verification, which may be insufficient to jointly capture token-level and sequenc-level confidencee.

The above two limitations motivate us to design a cloud-edge speculative decoding framework with high hardware resource utilization and flexible NAV triggering mechanisms to accelerate collaborative inference, which involves three main challenges:

**Complex pipeline scheduling considering cloud-edge communication startup overhead.** Intuitively, the communication of draft tokens can be overlapped with the autoregression of the draft model to improve resource utilization and to reduce inference latency, i.e., a draft token can be immediately transmitted once it is generated, as illustrated in Figure 2(b). However, the large startup overhead of communication mitigates the gains of this scheme. Therefore, standard pipeline patterns can hardly be applied.

**Joint impact of sequence-level and token-level confidence on NAV triggering.** Ideally, the triggering of NAV should be determined by the confidence of both entire draft sequence and individual tokens. Therefore, how to incorporate two factors to jointly determine NAV triggering and determine the optimal values of the two factors is challenging.

**Design of an adaptive and compatible collaborative inference framework.** In cloud-edge collaborative scenarios, the computing power of edge devices and the bandwidth of cloud-edge communication are usually unstable. An adaptive framework that can automatically adjust parameter configurations according to hardware environment changes is crucial. Furthermore, to ensure the compatibility of the designed framework with various cloud inference optimization

frameworks, optimization mechanisms should be deployed at the edge side to reduce dependency on the cloud.

In this paper, our main goal is to address the above three challenges by proposing an efficient cloud-edge collaborative pipeline inference framework with speculative decoding called PipeSD. All frequently used notations are summarized in Table A.1 of the Appendix.

# 3. PipeSD Design

## 3.1. Overview

Figure 3 illustrates the overall workflow of PipeSD within one speculative round, which includes four steps. Step 1: The edge device autoregressively generates draft tokens using a draft model. Step 2: While generating draft tokens, the edge batches and transmits them to overlap communication with computation (see details in Sec. 3.2), where an optimal batching strategy can be obtained by a DP algorithm (see details in Sec. 4.1). Step 3: The edge continuously evaluates both single-token and sequence confidence to determine whether to trigger NAV, and a lightweight BO autotuner is introduced to dynamically adjust the trigger thresholds (see details in Sec. 3.3). Step 4: Once NAV is requested, the cloud validates the accumulated draft tokens using a target model and returns the accepted tokens to the edge.

Steps 2 and 3 are the core steps of the proposed PipeSD. Step 2 effectively addresses the performance bottlenecks of low utilization of bandwidth and computing power by pipelining token generation and communication. Step 3 significantly enhances the flexibility of NAV triggering by introducing dual-threshold verification to simultaneously consider global and individual confidence.

## 3.2. Token-Batch Pipeline Scheduling Mechanism

We design an efficient token-batch pipeline scheduling mechanism that overlaps token autoregressive generation and communication, by deciding whether to merge draft tokens into one batch or transmit them immediately. For example, as shown in Figure 2(c), batching tokens 2 and 3 can further reduce total communication latency compared to immediate transmission. Therefore, we build a mathematical model to find the optimal token batching strategy that minimizes the autoregressive generation and communication time of draft tokens in a speculative round.

Given the number of draft tokens $N$ in each speculative round, we parameterize the batching strategy by a strictly increasing boundary sequence

$$\mathbb{B} = (b_1, \ldots, b_K), 1 = b_1 < b_2 < \cdots < b_K \le N \quad (1)$$

where $K$ is the number of token batches and $b_k$ is the index of the first draft token in the $k$-th token batch.

Then, the cloud-edge communication time of batch $k$, denoted by $t_c^{(k)}$, can be modeled as

$$t_c^{(k)} = \begin{cases} \alpha + \beta \cdot (b_{k+1} - b_k) & 1 \le k < K \\ \alpha + \beta \cdot (N + 1 - b_k) & k = K \end{cases} \quad (2)$$

where $\alpha$ is the startup overhead and $\beta$ is the per-token transmission time (see Sec. 5.2.4 for empirical validation and Appendix A for the modeling rationale). The autoregressive generation time of batch $k$, denoted by $t_{ag}^{(k)}$, can be represented as

$$t_{ag}^{(k)} = \begin{cases} \gamma \cdot (b_{k+1} - b_k) & 1 \le k < K \\ \gamma \cdot (N + 1 - b_k) & k = K \end{cases} \quad (3)$$

where $\gamma$ is the per-token computing time ($\gamma$ is assumed approximately constant within the optimization window, see Sec. 5.2.4 for empirical validation).

The autoregressive generation of a batch can start only after the previous batch's generation is complete. Thus the generation start time of batch $k$, denoted by $\tau_{ag}^{(k)}$, can be represented as:

$$\tau_{ag}^{(k)} = \begin{cases} 0 & k = 1 \\ \tau_{ag}^{(k-1)} + t_{ag}^{(k-1)} & k > 1 \end{cases} \quad (4)$$

Similarly, the communication of a batch can start only after both the previous batch's communication is completed and the current batch's generation is finished. Thus the communication start time of batch $k$, denoted by $\tau_c^{(k)}$, can be represented as:

$$\tau_c^{(k)} = \begin{cases} \tau_{ag}^{(k)} + t_{ag}^{(k)} & k = 1 \\ \max \left\{ \tau_c^{(k-1)} + t_c^{(k-1)}, \ \tau_{ag}^{(k)} + t_{ag}^{(k)} \right\} & k > 1 \end{cases} \quad (5)$$

Our goal is to find the optimal $\mathbb{B}$ that minimizes the autoregressive generation and communication time of draft tokens in a speculative round. The objective function and the dependencies can be expressed as follows:

$$\min_{\mathbb{B}} \quad T = \tau_c^{(K)} + t_c^{(K)} - \tau_{ag}^{(1)}$$
$$\text{s.t.} \quad Eqs. \ (1) - (5) \quad (6)$$

This optimization problem can be efficiently solved using a DP algorithm, which we describe in detail in Sec. 4.1.

## 3.3. Dual-threshold NAV Triggering Mechanism

We present a dual-threshold NAV triggering mechanism by jointly considering cumulative sequence confidence and single-token confidence. It precisely determines when to start the cloud NAV, avoiding both premature verification and excessive rollbacks.

For a draft token $D_n$, we denote its confidence as $P(D_n)$, i.e., the probability assigned by the draft model. Then, we use $C_1$ to denote the cumulative sequence confidence, which is the product of the confidence of all draft tokens that have

been computed but not yet verified, and $R_1$ to denote the cumulative sequence confidence threshold. When the draft model autoregressively generates the $n$-th draft token $D_n$, PipeSD updates the tentative cumulative confidence $C_1^*$ by multiplying $P(D_n)$ with the current $C_1$. If $C_1^* \leq R_1$, a cloud NAV is triggered, and $C_1$ is reset to 1. In addition, we use $R_2$ to denote the single-token confidence threshold. If $P(D_n) \leq R_2$, a cloud NAV is also triggered. Therefore, the proposed dual-threshold NAV triggering mechanism jointly considers both the confidence of the overall draft sequence and that of individual tokens, effectively improving the flexibility of NAV triggering.

It is worth noting that the threshold pair $(R_1, R_2)$ is closely related to the difficulty of the inference task, and has a significant impact on the speed of collaborative inference. Unfortunately, it is non-trivial to explicitly model the average generation time per accepted token (TPT) as a function of $(R_1, R_2)$. To ensure the adaptivity of PipeSD, we design a lightweight *BO autotuner* to automatically adjust $(R_1, R_2)$.

The BO autotuner aims to identify promising parameters of an unknown objective function using as few samples as possible. In our setting, the objective is to minimize the average TPT. The BO autotuner samples different triplets $(R_1, R_2, \text{TPT})$, and continuously suggests the next threshold pair $(R_1, R_2)$ to obtain a new objective value. The average TPT corresponding to each $(R_1, R_2)$ is measured by averaging the generation time of multiple accepted tokens. In other words, the BO autotuner efficiently predicts near-optimal thresholds after collecting only a small number of samples. In practical inference, with only 16 samples, the BO autotuner is able to return a near-optimal threshold pair. See Sec. 5.2.3 and Appendix C for more details on the performance evaluation and parameter settings of the BO autotuner.

Notably, under the dual-threshold triggering mechanism in PipeSD, the draft length $N$ of each speculative round is dynamically changing and implicitly determined by the confidence of the generated tokens. Thus, we introduce a scheduling window $\hat{N}$, where PipeSD performs pipeline scheduling of draft tokens in each speculative round with a granularity of $\hat{N}$ tokens. We dynamically adjust $\hat{N}$ based on the moving average length of the most recent 100 draft sequences ($\hat{N}$ is initialized to 20 in our experiments). Simultaneously, we establish two rules: (1) When a cloud NAV is triggered, the current pipeline scheduling period is interrupted and all unsent draft tokens are immediately transmitted in a single batch; (2) While awaiting cloud NAV results, the edge continues generating draft tokens and transmits them in batches with a period of $\hat{N}$ to further overlap computing and communication (see Appendix B for details).

**Algorithm 1** DP for Optimal Token Batching

---

**Input:** $\hat{N}, \alpha, \beta, \gamma$
1: **for** $j = 1$ **to** $\hat{N}$ **do**
2:    $\text{dp}[j] \leftarrow +\infty, \; \text{prev}[j] \leftarrow \varnothing$
3: $\text{dp}[0] \leftarrow 0$
4: **for** $j = 1$ **to** $\hat{N}$ **do**
5:    **for** $i = 0$ **to** $j - 1$ **do**
6:       $t_c \leftarrow \alpha + \beta \cdot (j - i)$        // *Eq. (2)*
7:       $temp \leftarrow \max\{\text{dp}[i], \gamma \cdot j\} + t_c$   // *Eqs. (3)-(5)*
8:       **if** $temp < \text{dp}[j]$ **then**
9:          $\text{dp}[j] \leftarrow temp, \; \text{prev}[j] \leftarrow i;$
10: *// Backtrack*
11: $\mathbb{B} \leftarrow (), \; p \leftarrow \hat{N}$
12: **while** $p > 0$ **do**
13:    $q \leftarrow \text{prev}[p], \; \mathbb{B} \leftarrow (q + 1, \mathbb{B}), \; p \leftarrow q$
14: **return** $\mathbb{B}$

---

## 4. Algorithm and System Implementation

### 4.1. Algorithm Design for Optimal Token Batching

We adopt a DP approach to obtain the optimal token batching strategy under the proposed pipeline model. Algorithm 1 summarizes the DP procedure.

The algorithm takes as input the scheduling window $\hat{N}$ and the communication and computation parameters ($\alpha$, $\beta$, $\gamma$), and returns the optimal batching strategy $\mathbb{B}$. First, it initializes a DP table $\text{dp}[\cdot]$, where $\text{dp}[j]$ represents the minimal autoregressive generation and communication time for the first $j$ tokens ($j \in [1, \hat{N}]$) and $\text{dp}[0] = 0$ is the base case for the empty prefix. Then, it iteratively fills the DP table by enumerating possible batch boundaries. Finally, $\mathbb{B}$ is recovered by backtracking through the DP table.

The time complexity of the DP algorithm is $O(\hat{N}^2)$, as it enumerates all pairs of batch boundaries $(i, j)$ with $i < j$. In practice, $\hat{N}$ is typically small, making the DP overhead negligible. Moreover, Algorithm 1 is re-executed to update the token batching strategy only when the scheduling window $\hat{N}$ changes or when the parameters ($\alpha$, $\beta$, $\gamma$) change substantially (see Appendix D.2 for details). Our experiments in Sec. 5.2.5 show that the DP overhead is less than $0.013\%$ of the total time of 1000 speculative rounds.

We also formally state the optimality of the DP-based batching algorithm (The proof is provided in Appendix E).

**Theorem 4.1.** *Under the pipeline model defined in Sec. 3.2, Algorithm 1 returns an optimal token batching strategy $\mathbb{B}$.*

### 4.2. System Implementation

Figure 4 illustrates the system architecture of PipeSD, which consists of an edge device and a cloud server. The edge device performs draft token generation, transmission control, and dynamic environment-aware adaptation, while the cloud only starts a FastAPI server and executes NAV using the

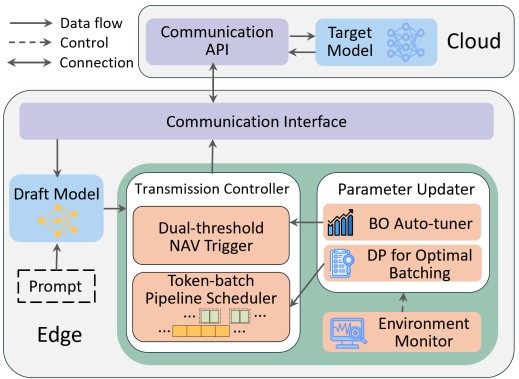

*Figure 4.* Overview of PipeSD architecture. The green part is the core of PipeSD.

target model, which makes the system easy to scale and compatible with existing cloud-edge collaborative frameworks. Moreover, although the current design only considers a single client, PipeSD can be easily extended to support multiple clients with minor modifications (see Appendix I for details).

The modules on the edge device include: (1) **Draft Model**, which generates draft tokens autoregressively, implemented with llama-cpp-python to enable efficient GGUF-based CPU inference (Hugging Face, 2023). Moreover, token-tree based drafting can also be enabled in PipeSD (see Appendix J for details); (2) **Transmission Controller**, which orchestrates pipeline scheduling and NAV triggering by *Token-batch Pipeline Scheduler* and *Dual-threshold NAV Trigger*, respectively; (3) **Communication Interface**, which uploads draft tokens and NAV requests to the cloud server, and receives NAV results; (4) **Environment Monitor**, which continuously monitors the average TPT and the communication and computation parameters $(\alpha, \beta, \gamma)$, and triggers targeted updates when significant changes are detected(see Appendix D for details); and (5) **Parameter Updater**, which re-runs the BO autotuner to update $(R_1, R_2)$ upon significant TPT changes, and re-executes the DP scheduler (Algorithm 1) to update $\mathbb{B}$ upon significant changes in $(\alpha, \beta, \gamma)$ (see Appendix D for details).

The modules on the cloud include: (1) **Communication API**, which starts a FastAPI server and exposes endpoints for receiving draft tokens and NAV signals from the edge device, and returning NAV results; and (2) **Target Model**, which performs NAV on the received draft tokens.

## 5. Evaluation

### 5.1. Experimental Setups

**Testbed.** Our cloud-edge testbed is deployed in a real-world metropolitan network environment. The **edge device** is a Lenovo ThinkBook 16+ equipped with an Intel® Core™ Ultra 9 185H CPU (16 cores, up to 5.1 GHz) and 32 GB of system memory, running Windows 11 (24H2). The **cloud server** is hosted on Tianyi Cloud and equipped with an NVIDIA A800 GPU (40 GB VRAM), an Intel Xeon CPU, and 120 GB of system memory, running Ubuntu 22.04 LTS. The uplink and downlink bandwidths are 20 Mbps and 200 Mbps, respectively, which meet the standard 5G communication bandwidth (Wu et al., 2024). We construct four experimental scenarios. Scenarios 1–3 use the above static network setting but differ in the edge device compute capability. Specifically, Scenario 1 uses the above-mentioned Lenovo ThinkBook as the edge device, while Scenarios 2 and 3 emulate a mobile phone and an IoT device, respectively. We set the computing power of the simulated mobile phone and IoT device to 2.5 GHz and 1.2 GHz, respectively, which are common device frequencies (Liu et al., 2024; Fitzgibbon & Ottaviani, 2024). We emulate these devices by adding corresponding latency on the same testbed (see Appendix G.2 for emulation details). Scenario 4 uses the same edge device setting as Scenario 1 but applies a dynamic-bandwidth setting. In this scenario, the uplink and downlink bandwidths vary within $[10, 80]$ Mbps and $[150, 280]$ Mbps, respectively, with a change interval of 20 seconds (Al-Falahy & Alani, 2017) (see Appendix G.1 for bandwidth control details).

**Models and Datasets.** We evaluate PipeSD on programming and mathematical reasoning tasks. For programming, we use the HumanEval dataset (Chen et al., 2021), with DeepSeek-Coder-1.3B as the draft model and DeepSeek-Coder-6.7B as the target model (Guo et al., 2024). For mathematical reasoning, we use GSM8K (Cobbe et al., 2021), where TinyLlama-1.1B-Chat-v1.0 (Zhang et al., 2024) and Llama-2-7B (Touvron et al., 2023b) serve as the draft and target models, respectively.

**Baselines.** Some pioneering works are orthogonal to PipeSD: HAT studies cloud-edge collaborative speculative decoding under relaxed accuracy constraints, while SpecEdge focuses on multi-edge collaboration. We benchmark PipeSD against the following representative frameworks: (1) Vanilla Cloud-Edge Collaborative Speculative Decoding (Vanilla) (Kim et al., 2023), where the edge device autoregressively generates and uploads a fixed number of draft tokens, we set $N = 6$ for programming tasks and $N = 4$ for mathematical reasoning tasks, which yield the best performance across all scenarios. (2) HSL (Hao et al., 2024), which triggers cloud NAV when the confidence of a draft token falls below a predefined threshold. We set the thresholds to 0.99 for programming tasks and 0.7 for mathematical reasoning tasks for HSL's best performance across all scenarios. (3) EdgeLLM (Xu et al., 2025), which is adapted for the cloud-edge collaboration scenario while retaining its two key mechanisms: (i) continuing draft gener-

*Table 1.* Comparison of average TPT (ms). **ID** denotes the scenario index. $S_{t1}$, $S_{t2}$, and $S_{t3}$ denote the speedup of PipeSD over Vanilla, HSL, and EdgeLLM, respectively.

| ID | Dataset | TPT (ms) | | | | Speedup | | |
|---|---|---|---|---|---|---|---|---|
| | | Vanilla | HSL | EdgeLLM | PipeSD | $S_{t1}$ | $S_{t2}$ | $S_{t3}$ |
| 1 | HumanEval | 194 | 155 | 153 | 129 | 1.50× | 1.20× | 1.19× |
| | GSM8K | 193 | 174 | 169 | 145 | 1.33× | 1.20× | 1.17× |
| 2 | HumanEval | 225 | 184 | 166 | 134 | 1.68× | 1.37× | 1.24× |
| | GSM8K | 318 | 223 | 197 | 168 | 1.89× | 1.33× | 1.17× |
| 3 | HumanEval | 306 | 244 | 201 | 152 | 2.01× | 1.61× | 1.32× |
| | GSM8K | 402 | 296 | 231 | 186 | 2.16× | 1.59× | 1.24× |
| 4 | HumanEval | 160 | 132 | 127 | 108 | 1.48× | 1.22× | 1.18× |
| | GSM8K | 234 | 165 | 161 | 139 | 1.68× | 1.19× | 1.16× |

*Table 2.* Comparison of ECS (J) in Scenario 1. $P_{e1}$, $P_{e2}$, and $P_{e3}$ denote the ECS reduction of PipeSD compared to Vanilla, HSL, and EdgeLLM, respectively.

| Dataset | ECS (J) | | | | ECS Reduction (%) | | |
|---|---|---|---|---|---|---|---|
| | Vanilla | HSL | EdgeLLM | PipeSD | $P_{e1}$ | $P_{e2}$ | $P_{e3}$ |
| HumanEval | 68 | 71 | 75 | 56 | 17.6 | 21.1 | 25.3 |
| GSM8K | 98 | 102 | 100 | 84 | 14.3 | 17.6 | 16.0 |

ation while waiting for NAV, and (ii) triggering NAV when the cumulative sequence confidence falls below a dynamically adjusted threshold (see Appendix G.3 for details).

**Metrics.** We focus on two metrics: (i) the average generation time per accepted token (TPT), and (ii) the average energy consumption of the cloud server per 100 accepted tokens (ECS). All reported results are averaged over 1000 accepted tokens.

### 5.2. Experimental Results

#### 5.2.1. OVERALL PERFORMANCE COMPARISON

**TPT Comparison:** Table 1 reports the average TPT of all methods on both datasets across four scenarios. The results show that PipeSD achieves speedups of 1.33×–2.16×, 1.19×–1.61×, and 1.16×–1.32× compared to Vanilla, HSL, and EdgeLLM, respectively. For Scenarios 2–3 with limited computing power of the edge, PipeSD attains larger gains because more communication time can be hidden by token-batch pipelining. Under dynamic bandwidth (Scenario 4), PipeSD still achieves consistent improvements, demonstrating its robustness to bandwidth fluctuations (see Appendix D for more details). Overall, performance improvement mainly comes from two mechanisms of PipeSD: token-batch pipeline scheduling overlaps computing and communication to hide communication latency, while dual-threshold NAV triggering enables timely verification, avoiding both premature and delayed triggering. **ECS Comparison:** We compute ECS by time-integrating the cloud

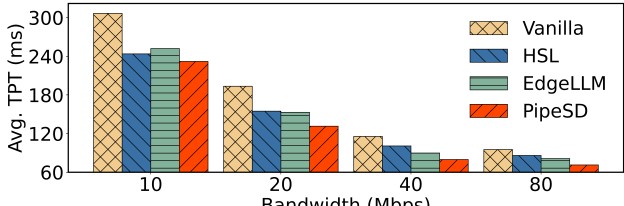

*Figure 5.* Average TPT (ms) with different bandwidth levels on HumanEval in Scenario 1.

*Table 3.* Performance evaluation of BO autotuner in Scenario 1.

| Dataset | TPT (ms) | | |
|---|---|---|---|
| | BO Autotuner | Grid Search | Random Search |
| HumanEval | 129 | 139 | 148 |
| GSM8K | 145 | 155 | 162 |

GPU power trace sampled with NVIDIA SMI (NVIDIA Corporation, 2025) at 5 ms intervals (Gao et al., 2025). Table 2 shows the ECS of all methods on both datasets in Scenario 1. Compared to Vanilla, HSL, and EdgeLLM, PipeSD achieves ECS reductions of 17.6%, 21.1%, and 25.3% on HumanEval, and 14.3%, 17.6%, and 16.0% on GSM8K, respectively. This improvement is mainly attributed to the more accurate and efficient verification triggering brought by the dual-threshold NAV triggering mechanism, which effectively reduces unnecessary verification requests. While ECS captures cloud-side energy, it is also important to assess whether PipeSD incurs additional energy overhead on the edge. However, accurately measuring edge-side energy is challenging, because the edge runs inference on a general-purpose CPU and the measured power can be affected by background OS activities and other software processes. We therefore provide a theoretical analysis of the edge-side energy consumption in Appendix H, which indicates that PipeSD introduces only negligible additional edge energy overhead from BO autotuning, DP scheduling, and parameter measurement.

#### 5.2.2. IMPACT OF NETWORK BANDWIDTH

Figure 5 shows the performance of PipeSD under different uplink bandwidths on HumanEval in Scenario 1. At 10, 20, 40, and 80 Mbps, PipeSD accelerates inference by 1.32×, 1.47×, 1.45×, and 1.34× over Vanilla; 1.05×, 1.18×, 1.26×, and 1.21× over HSL; and 1.09×, 1.16×, 1.13×, and 1.14× over EdgeLLM, respectively. We observe that the average TPT stabilizes as the bandwidth reaches 80 Mbps. This is because, at higher bandwidth levels, communication is no longer the performance bottleneck.

*Table 4.* Comparison of TPT (ms) when using PipeSD with BO autotuner or different fixed $(R_1, R_2)$ on HumanEval in Scenario 1.

| TPT (ms) | | | | | | | | | |
|---|---|---|---|---|---|---|---|---|---|
| BO | (0.3, 0.3) | (0.3, 0.6) | (0.3, 0.9) | (0.6, 0.3) | (0.6, 0.6) | (0.6, 0.9) | (0.9, 0.3) | (0.9, 0.6) | (0.9, 0.9) |
| 129 | 197 | 174 | 139 | 189 | 174 | 139 | 149 | 156 | 138 |

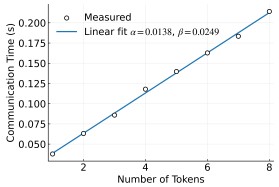

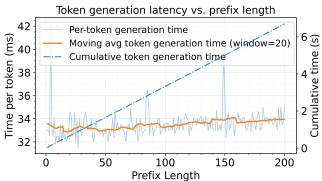

*(a)* Communication time vs. number of tokens in a batch. *(b)* Per token generation time $\gamma$ with respect to the prefix length.

*Figure 6.* Communication and computation latency characteristics used in PipeSD.

### 5.2.3. PERFORMANCE EVALUATION OF BO AUTOTUNER

We evaluate the effectiveness of BO autotuner by comparing it with grid search and random search on tuning $(R_1, R_2)$. As shown in Table 3, BO autotuner consistently achieves the lowest average TPT on both HumanEval and GSM8K. Compared with grid search and random search, BO is able to efficiently converge to near-optimal configurations with minimal overhead, making it more suitable for online parameter optimization (see Appendix C.2 for more analysis).

In addition, we compare the average TPT when using PipeSD with BO autotuner or fixed $(R_1, R_2)$ pairs on HumanEval in Scenario 1. As shown in Table 4, different $(R_1, R_2)$ greatly affect the inference efficiency and BO auto-tuning is essential to maximize the performance of PipeSD.

### 5.2.4. PARAMETER MEASUREMENT

**Measurement for $\alpha$ and $\beta$:** We measure the communication latency for transmitting token batches of different sizes. As shown in Figure 6a, the communication time increases linearly with the number of tokens in a batch, where the intercept and slope of the fitted line correspond to $\alpha$ and $\beta$, respectively. **Measurement for $\gamma$:** Figure 6b shows that, within a 200-token sliding window, $\gamma$ remains approximately constant across different prefix lengths.

### 5.2.5. OVERHEAD ANALYSIS

We measure the computational overhead of the BO auto-tuner, DP scheduler and parameter measurement by profiling their runtime during the first 1000 speculative rounds on both datasets in Scenario 1. Table 5 reports their overhead as a percentage of the total execution time. The results show that the overhead of the BO autotuner is negligible,

*Table 5.* Percentage of the overhead of BO autotuner, DP scheduler and parameter measurement to the total time of first 1000 speculative rounds in Scenario 1.

| Dataset | HumanEval | GSM8K |
|---|---|---|
| Overhead of BO autotuner | 1.1% | 0.9% |
| Overhead of DP scheduler | 0.01% | 0.013% |
| Overhead of parameter measurement | 0.3% | 0.4% |

*Table 6.* Ablation studies on HumanEval in Scenario 1.

| Method | Pipeline | NAV trigger | TPT (ms) | Speedup |
|---|---|---|---|---|
| Vanilla | ✗ | Fixed-length | 194 | 1.00× |
| PipeSD w/o Pipeline | ✗ | Dual-threshold | 147 | 1.32× |
| PipeSD + Fixed-length | ✓ | Fixed-length | 164 | 1.18× |
| PipeSD + Token-level | ✓ | Token-level | 137 | 1.42× |
| PipeSD + Sequence-level | ✓ | Sequence-level | 139 | 1.40× |
| PipeSD (Full) | ✓ | Dual-threshold | 129 | 1.50× |

accounting for no more than 1.1% on both datasets, which underscores its efficiency as a parameter-tuning mechanism. In addition, the overhead of parameter measurement contributes less than 0.4%, as it is executed only when significant environmental changes are detected. The overhead of the DP scheduler is negligible (below 0.013%).

### 5.2.6. ABLATION STUDIES

We conduct ablation studies on HumanEval in Scenario 1. The results are shown in Table 6, where (1) *PipeSD w/o Pipeline* disables the pipeline scheduling and instead transmits draft tokens after generating the entire draft sequence, (2) *PipeSD + Fixed-length* adopts a fixed-length NAV triggering strategy, (3) *PipeSD + Token-level* uses a single-token confidence-based strategy, and (4) *PipeSD + Sequence-level* adopts a cumulative sequence confidence-based strategy. It is seen that *PipeSD w/o Pipeline* is 1.12× slower than the full PipeSD, which demonstrates the effectiveness of token-batch pipeline scheduling mechanism. Meanwhile, compared to *PipeSD + Fixed-length*, *PipeSD + Token-level*, and *PipeSD + Sequence-level*, the full PipeSD achieves speedups of 1.25×, 1.05×, and 1.06×, respectively, which indicates that jointly considering both single-token and cumulative sequence confidence provides more effective NAV triggering.

To further verify that the benefit of the pipeline mechanism does not merely come from introducing pipelining, but also from the proposed DP-based token-batching policy, we compare the DP-based token-batching policy with stronger pipelined baselines, including a greedy policy that sends all accumulated tokens whenever the network becomes idle, as well as two heuristic policies, namely immediate-send and

*Table 7.* Speculative-decoding statistics on HumanEval in Scenario 1.

| Method | Verification Frequency | Mean Draft Length | Acceptance Rate |
|---|---|---|---|
| HSL | 0.2558 | 3.18 | 0.9148 |
| EdgeLLM | 0.1912 | 4.74 | 0.8917 |
| PipeSD | 0.1733 | 4.96 | 0.9616 |

no-early-upload. The detailed setup and results are reported in Appendix F, where DP consistently achieves the best performance with negligible scheduling overhead.

To further understand the advantage of the dual-threshold NAV triggering mechanism, we additionally report several speculative-decoding statistics on HumanEval in Scenario 1, which provide a more fine-grained view of verification behavior beyond TPT alone. Table 7 provides a more detailed view of the NAV-triggering behavior of different methods. HSL is overly conservative with frequent verification and short draft sequences, limiting speculative gains. EdgeLLM is more balanced, but still achieves a lower acceptance rate than PipeSD. PipeSD achieves the best trade-off by maintaining a moderate draft length while attaining the highest acceptance rate. This suggests that the dual-threshold NAV triggering mechanism enables more accurate verification decisions, thereby reducing end-to-end inference latency.

## 6. Conclusion

In this paper, we propose PipeSD, an efficient cloud-edge collaborative inference framework with speculative decoding. First, PipeSD introduces a token-batch pipeline scheduling mechanism that overlaps draft token generation and communication, and leverages a DP algorithm to obtain optimal token batching strategies, thereby improving overall resource utilization. Second, PipeSD employs a dual-threshold NAV triggering mechanism to enhance verification flexibility, and incorporates a lightweight BO autotuner to automatically adjust thresholds. We implement PipeSD using llama-cpp-python, PyTorch, and FastAPI, and evaluate it on real-world cloud-edge testbeds. Extensive experiments with two draft-target model pairs across four scenarios demonstrate the superiority of PipeSD over state-of-the-art baselines, including HSL and EdgeLLM, achieving $1.16\times$–$2.16\times$ speedup and reducing energy consumption by 14.3%–25.3%. Future work will validate PipeSD's robustness in more diverse real-world settings, including varying network conditions, hardware platforms, and task types.

## Acknowledgements

This work was supported in part by the Zhongguancun Academy, (Grant No.s XTS0038), and in part by Information Technology Center and State Key Lab of CAD&CG, ZheJiang University.

## Impact Statement

This paper presents work whose goal is to advance the field of Machine Learning. There are many potential societal consequences of our work, none which we feel must be specifically highlighted here.

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

*Table A.1.* Frequently used notations.

| Symbol | Description |
| --- | --- |
| $N$ | Number of draft tokens generated in one speculative round |
| $\hat{N}$ | Token-batch scheduling window |
| $K$ | Number of token batches in a batching strategy |
| $\mathbb{B}$ | Token batching strategy, represented as a set of batch start indices |
| $b_k$ | Index of the first draft token in the $k$-th batch |
| $t_c^{(k)}$ | Cloud-edge communication time of the $k$-th token batch |
| $t_{ag}^{(k)}$ | Autoregressive generation time of the $k$-th token batch |
| $\tau_{ag}^{(k)}$ | Start time of autoregressive generation for the $k$-th batch |
| $\tau_c^{(k)}$ | Start time of communication for the $k$-th batch |
| $\alpha$ | Cloud-edge communication startup overhead |
| $\beta$ | Per-token transmission time |
| $\gamma$ | Per-token autoregressive generation time on the edge device |
| $T$ | Total generation and communication time of draft tokens in one speculative round |
| $D_n$ | The $n$-th draft token generated by the draft model |
| $P(D_n)$ | Confidence of draft token $D_n$ |
| $C_1$ | Cumulative sequence confidence of draft tokens |
| $C_1^*$ | Updated cumulative sequence confidence after generating a new token |
| $R_1$ | Threshold for cumulative sequence confidence triggering NAV |
| $R_2$ | Threshold for single-token confidence triggering NAV |
| NAV | Non-autoregressive verification performed by the target model |
| TPT | Generation time per accepted token |
| ECS | Energy consumption of the cloud server per 100 accepted tokens |

## A. Modeling Rationale for Communication Time

In addition to empirical validation, we provide a theoretical rationale for our model. Specifically, the model is based on the widely adopted linear communication time model proposed by Hockney (Hockney, 1994). This model characterizes communication time using two key components: (1) Startup Overhead ($\alpha$): This term accounts for the fixed latency incurred when initiating a communication session between the edge device and the cloud server (Gao et al., 2024). This overhead includes factors such as network handshake, connection establishment, and protocol negotiation, which are independent of the size of the data being transmitted. (2) Data Transmission Time ($\beta \cdot n$): This term represents the variable component of communication time that scales with the number of tokens ($n$) being transmitted in the batch. Here, $\beta$ is the average time taken to transmit a single token over the network, which depends on factors such as bandwidth, network congestion, and packet size.

## B. Proactive Transmission of Draft Tokens

To further reduce the edge-side idle time when waiting for verification result, PipeSD proactively generates and transmits draft tokens while verification is still in progress. Specifically, after sending the last batch of a speculative round, the edge immediately starts generating and transmitting the subsequent token batches without waiting for the verification result. The cloud buffers these proactively sent draft tokens upon arrival. After finishing NAV for the current round, the cloud checks whether (i) all draft tokens are accepted and (ii) the extra token generated by the target model matches the first buffered draft token. If both conditions hold, the buffered tokens remain valid and are kept; otherwise, they are discarded. Upon receiving the verification result, the edge applies the same check: if the draft tokens of the current round are fully accepted and the extra token given by the cloud matches the first proactively generated draft token, it resumes draft generation from where it left off; otherwise, it restarts draft generation from the last accepted token.

## C. Design and Performance Evaluation of BO

### C.1. Parameter Settings and Advantage Analysis

We choose BO to obtain a near-optimal $(R_1, R_2)$ benefiting from the following three points:

- First, BO is not limited by the expression of the objective function (we use $\mathcal{F}(R_1, R_2)$ as the objective function) and depends only on the sampling values obtained (i.e., $\hat{\mathcal{F}}(R_1^1, R_2^1), \hat{\mathcal{F}}(R_1^2, R_2^2), \ldots, \hat{\mathcal{F}}(R_1^n, R_2^n)$). We use Gaussian process regression with the Matern kernel to predict the value of the objective function as it is commonly used as a good surrogate model for BO (Alipourfard et al., 2017).

- Second, BO typically requires only a limited number of trials to find high-quality solutions, resulting in low search overhead. To minimize the number of trials, it selects the next configuration $(R_1^1, R_2^1)$ by maximizing an acquisition function (Alipourfard et al., 2017). In PipeSD, we adopt the Expected Improvement (EI) acquisition function to choose the $(R_1, R_2)$.

- Third, BO mitigates the risk of converging to a local optimum by adjusting the hyperparameter Expected Improvement (EI). Specifically, a smaller EI encourages exploitation by sampling more densely near the current optimum, whereas a larger EI promotes exploration by selecting more diverse points across the search space (Alipourfard et al., 2017). In PipeSD, we set EI $= 0.1$ to favor exploration over $(R_1, R_2)$, which is a commonly adopted setting. Meanwhile, the search space of $(R_1, R_2)$ in BO is defined as $(0, 1)^2$, and a single initial sample is randomly generated to initialize the optimization process.

### C.2. Detailed Comparative Analysis of Tuning Strategies

To further justify the efficiency of the BO autotuner, we detail the implementation of the baseline search methods. In our evaluation, all methods share the same search space $(0, 1)^2$ for $(R_1, R_2)$.

- **Grid Search:** The search space is discretized into a $4 \times 4$ uniform grid, resulting in 16 deterministic sampling points.

- **Random Search:** 16 pairs of $(R_1, R_2)$ are sampled independently and uniformly from the continuous search space.

Evaluation Protocol: Each sampling point is evaluated by measuring the average TPT over 20 accepted tokens to ensure statistical stability and mitigate measurement noise. Table 3 shows the comparison of the average TPT when using different tuning methods on HumanEval and GSM8K. The results show that BO autotuner achieves the lowest TPT on both datasets. Although grid search obtains better performance than random number generation, its limited number of sampling points makes it difficult to cover the optimal solution especially when the search space is large, and over-increasing the number of sampling points brings higher search overhead and longer search process. On the contrary, BO can obtain a near-optimal solution with very little overhead, and therefore we choose it.

## D. Robustness of PipeSD

In real-world deployment, the hardware environments including network bandwidth and computing power of the edge often change dynamically. In addition, task difficulty varies across inputs, which can shift the optimal NAV-triggering thresholds. To adapt to such dynamics, PipeSD continuously monitors online perfomance metrics and triggers targeted updates: (i) when the average TPT changes significantly, it re-runs the BO autotuner to update the NAV thresholds $(R_1, R_2)$, and (ii) when the communication and computation parameters $(\alpha, \beta, \gamma)$ change significantly, it re-executes the DP scheduler to update the token batching strategy with updated parameters.

### D.1. Automatic Update of NAV Thresholds

When average TPT changes noticeably, we re-run the BO autotuner to obtain new $(R_1, R_2)$. We monitor TPT using a sliding window over the most recent 100 accepted tokens. An update is triggered only after the window is full and the relative change in TPT exceeds a predefined threshold $\delta_1$:

$$\frac{|\text{TPT}_{\text{new}} - \text{TPT}_{\text{old}}|}{\text{TPT}_{\text{old}}} > \delta_1.$$

Here, $\text{TPT}_{\text{new}}$ and $\text{TPT}_{\text{old}}$ denote the average TPTs of the current and previous windows, respectively. The threshold $\delta_1$ depends on the hardware environment and inference configuration. When the condition is met, the BO autotuner is re-executed asynchronously.

### D.2. Automatic Update of Token Batching Strategy

When the communication and computation parameters ($\alpha$, $\beta$, and $\gamma$) change significantly, we update the token batching strategy by re-running the DP scheduler (Algorithm 1) with updated parameters. We estimate these parameters using a sliding window over the most recent 100 transmitted token batches, as described below.

- **Estimating $\gamma$.** We compute $\gamma$ as the average *per-token* generation time over the most recent 100 generated batches (or all available batches if fewer than 100 are available).

- **Estimating $\alpha$ and $\beta$.** We bootstrap the estimation by transmitting 8 token batches with sizes 1–8 and recording their end-to-end communication times, which provides the initial data points for regression. Afterward, we maintain a sliding window over the most recent 100 transmitted batches; the estimation below is performed only once the window is full. For each batch in the window, we record its end-to-end communication time, group batches by size, and compute the average communication time per size. If fewer than 8 distinct batch sizes appear in the window, we proactively transmit additional batches with previously unseen sizes (starting from the smallest unseen sizes) to obtain up to 8 data points. We then fit a linear model of average communication time versus batch size, where the intercept and slope correspond to $\alpha$ and $\beta$, respectively.

To avoid unnecessary re-scheduling, we trigger a DP update only when the estimated parameters change noticeably. Specifically, we re-run the DP scheduler if the relative change in $\gamma$ exceeds $\delta_2$, or if the relative change in either $\alpha$ or $\beta$ exceeds a shared threshold $\delta_3$:

$$\frac{|\gamma_{\text{new}} - \gamma_{\text{old}}|}{\gamma_{\text{old}}} > \delta_2 \quad \text{or} \quad \frac{|\alpha_{\text{new}} - \alpha_{\text{old}}|}{\alpha_{\text{old}}} > \delta_3 \quad \text{or} \quad \frac{|\beta_{\text{new}} - \beta_{\text{old}}|}{\beta_{\text{old}}} > \delta_3.$$

Here, $(\alpha_{\text{new}}, \beta_{\text{new}}, \gamma_{\text{new}})$ and $(\alpha_{\text{old}}, \beta_{\text{old}}, \gamma_{\text{old}})$ are the parameter estimates from the current and previous windows, respectively. The thresholds $\delta_2$ and $\delta_3$ depend on the hardware environment and inference configuration. When either condition holds, we update the token batching strategy by re-executing the DP scheduler.

If the thresholds are set too small, measurement noise may frequently trigger unnecessary updates, resulting in excessive overhead. In contrast, overly large thresholds may delay the detection of meaningful environmental changes, reducing the adaptivity of PipeSD.

In our experiments, we empirically set

$$\delta_1 = \delta_2 = \delta_3 = 0.2,$$

which provides a good balance between adaptivity and update overhead. We further observe that PipeSD is not highly sensitive to the exact threshold values within a moderate range. Specifically, when the thresholds vary within $[0.1, 0.5]$, the overall inference performance remains very similar across all evaluated scenarios. This empirical observation demonstrates the robustness of PipeSD with respect to threshold selection.

## E. Proof of Theorem 4.1

For any $j \in \{0, 1, \ldots, \hat{N}\}$, define $\text{OPT}(j)$ as the minimum possible autoregressive generation and communication completion time of all tokens $\{1, \ldots, j\}$. Clearly, $\text{OPT}(0) = 0$.

Consider an optimal strategy for the prefix $\{1, \ldots, j\}$ with $j \geq 1$. Let the last batch be $\{i + 1, \ldots, j\}$ for some $i \in \{0, \ldots, j-1\}$. Its communication time is $\alpha + \beta (j - i)$. The communication of this last batch can start only after both the previous batch has completed communication and the current batch has completed autoregressive generation. The former completes at time $\text{OPT}(i)$ by definition. Our experimental results (Figure 6b) show that the per-token generation time on the edge device remains approximately constant with respect to the prefix length. We denote this constant generation time by $\gamma$. Consequently, generating $j$ tokens requires approximately $\gamma j$ time, and the latter condition is satisfied at time $\gamma j$.

*Table A.2.* Speedup of DP-based token batching over stronger pipelined baselines under different communication settings.

| $(\alpha, \beta)$ (ms) | DP vs Greedy | DP vs Immediate-send | DP vs No-early-upload |
|---|---|---|---|
| $(20, 72)$ | $1.02\times$ | $1.09\times$ | $1.22\times$ |
| $(100, 72)$ | $1.04\times$ | $1.44\times$ | $1.10\times$ |
| $(200, 72)$ | $1.10\times$ | $1.76\times$ | $1.06\times$ |
| $(20, 48)$ | $1.03\times$ | $1.11\times$ | $1.33\times$ |
| $(100, 48)$ | $1.06\times$ | $1.63\times$ | $1.13\times$ |
| $(200, 48)$ | $1.13\times$ | $2.06\times$ | $1.06\times$ |

Therefore, the earliest possible communication start time of the last batch is

$$\max\{\mathrm{OPT}(i),\ \gamma j\},$$

and the resulting completion time equals

$$\max\{\mathrm{OPT}(i),\ \gamma j\} + \alpha + \beta\,(j - i).$$

Minimizing over all possible $i$ yields the following recurrence:

$$\mathrm{OPT}(j) = \min_{0 \leq i \leq j-1} \left(\max\{\mathrm{OPT}(i),\ \gamma j\} + \alpha + \beta\,(j - i)\right). \tag{7}$$

The DP in Algorithm 1 computes $\mathrm{dp}[j]$ using exactly the recurrence in (7) with base $\mathrm{dp}[0] = 0$, hence by induction on $j$ we have $\mathrm{dp}[j] = \mathrm{OPT}(j)$ for all $j \leq \hat{N}$. In particular, $\mathrm{dp}[\hat{N}] = \mathrm{OPT}(\hat{N})$, so the minimum completion time is achieved.

Finally, storing a predecessor for each $j$ and backtracking reconstructs a partition whose boundaries attain the minimum in (7) at every step, and thus yields an optimal batching strategy $\mathbb{B}$.

Therefore, Algorithm 1 returns an optimal token batching strategy.

## F. Necessity of DP-based Token Batching

To further justify the necessity of the DP-based token-batching policy, we compare PipeSD with several stronger pipelined baselines beyond the no-pipeline ablation in Table 6. The goal of this comparison is to isolate whether the gain of PipeSD comes merely from introducing pipelining, or from the DP-based optimization of token-batch transmission itself.

We consider the following baselines:

- **Greedy:** whenever the network becomes idle, the edge immediately transmits all currently accumulated draft tokens.

- **Immediate-send:** each draft token is transmitted as soon as it is generated.

- **No-early-upload:** the edge first generates the whole draft sequence and then uploads it to the cloud, i.e., no transmission pipelining is applied.

We evaluate these methods under different communication settings characterized by the startup latency $\alpha$ and per-token transmission latency $\beta$. Table A.2 reports the speedup of DP over the above baselines.

The results show that DP consistently outperforms all three alternatives. In particular, compared with the greedy policy, DP still achieves $1.02\times$–$1.13\times$ speedup across all tested settings. This indicates that the advantage of PipeSD does not merely come from pipelining itself, but from optimizing *when* and *how many* tokens should be transmitted under communication startup overhead. The gain over immediate-send is even larger, especially when $\alpha$ is high, because transmitting too frequently incurs excessive startup cost. Meanwhile, DP also consistently outperforms no-early-upload, confirming the benefit of overlapping token generation with communication.

These results, together with the negligible DP overhead reported in Table 5, demonstrate that the DP scheduler is both necessary and practical in PipeSD.

## G. Details of Experimental Setup

### G.1. Network Bandwidth Control

To ensure reproducible and fair comparisons, we conduct all experiments under fixed network bandwidth settings, except where explicitly noted. We limit the bandwidth on the cloud-edge link using OS-specific traffic-control tools on each endpoint. The edge device runs Windows 11; we limit the uplink bandwidth using Windows Policy-based QoS by configuring an outbound throttling rate for the traffic from the edge to the cloud. The cloud server runs Ubuntu 22.04; we limit the downlink bandwidth using Linux Traffic Control (`tc`) (Hubert, 2001).

### G.2. Edge Compute Emulation with Artificial Delays

To emulate Scenario 2 and Scenario 3 with limited computing power of the edge device, we inject artificial delays into token generation on the edge device. Our physical edge device (Lenovo ThinkBook 16+) has a CPU frequency of 5.1 GHz. We simulate lower CPU frequencies of 2.5 GHz (Scenario 2, smartphone-class) and 1.2 GHz (Scenario 3, IoT-class) by adding an extra delay after each generated token. The per-token artificial delay is computed as

$$\text{Artificial Delay} = \text{Base Generation Time} \times \left( \frac{\text{Real CPU Frequency}}{\text{Simulated CPU Frequency}} - 1 \right),$$

where Base Generation Time denotes the time to generate one token on the physical edge device.

### G.3. EdgeLLM Adaptation Details

EdgeLLM is originally designed for edge-only inference scenarios, but its two core ideas transfer naturally to our cloud-edge collaborative setting: (i) continuing draft generation while waiting for NAV, and (ii) triggering NAV when the cumulative sequence confidence falls below a dynamically adjusted cumulative sequence confidence threshold $R_1$.

The first technique is implemented in the same way as our proactive transmission of draft tokens (Appendix B). Below we describe EdgeLLM's dynamic threshold mechanism. Consider a speculative round with a scheduling window of $\hat{N}$ draft tokens. After each NAV, we update $R_1$ using the following rule:

$$R_{1,\text{new}} = \begin{cases} 0.5\, R_1, & \text{if } N_{\text{correct}} = \hat{N}, \\ \dfrac{R_1}{C_1^{\frac{\hat{N} - N_{\text{correct}}}{\hat{N}}}}, & \text{if } N_{\text{correct}} < \hat{N}. \end{cases} \tag{7}$$

where $N_{\text{correct}}$ is the number of accepted tokens in the current round, and $C_1$ is the cumulative sequence confidence before NAV. In our experiments, we choose the optimal initial value of $R_1$ for each experimental setup.

## H. Edge Energy Overhead

We analyze the additional energy consumed on the edge due to PipeSD's control-plane logic (BO autotuning, DP scheduling, and online parameter measurement). Let $P_{\text{idle}}$ denote the CPU idle power on the edge device. When the edge executes a workload $x$, its average power is modeled as $P_{\text{idle}} + P_x$, where $P_x$ is the workload-induced incremental power above idle (Hähnel et al., 2012; Tiwari et al., 1994). In particular, let $P_{\text{ag}}$ be the incremental power of autoregressive token generation, and let $P_{\text{BO}}$, $P_{\text{DP}}$, and $P_{\text{PM}}$ be the incremental powers of BO, DP, and parameter measurement, respectively. Let $T_{\text{BO}}$, $T_{\text{DP}}$, and $T_{\text{PM}}$ be their average per-round runtimes, and let $T$ be the average duration of one speculative decoding round. Note that BO autotuning, DP scheduling, and parameter measurement are *not* executed in every speculative round; they are triggered only when necessary. Therefore, in the per-round analysis, the terms $T_{\text{BO}}$, $T_{\text{DP}}$, and $T_{\text{PM}}$ should be interpreted as *amortized* runtimes, i.e., the total time spent on these routines over many rounds divided by the number of rounds.

We consider a conservative (worst-case) setting where these control-plane steps are *not* overlapped with draft generation. Then the edge spends $T - T_{\text{BO}} - T_{\text{DP}} - T_{\text{PM}}$ on draft generation. The total edge energy per round can be written as

$$\begin{aligned} W &= (T - T_{\text{BO}} - T_{\text{DP}} - T_{\text{PM}})\,(P_{\text{ag}} + P_{\text{idle}}) \\ &\quad + T_{\text{DP}}(P_{\text{DP}} + P_{\text{idle}}) + T_{\text{PM}}(P_{\text{PM}} + P_{\text{idle}}) + T_{\text{BO}}(P_{\text{BO}} + P_{\text{idle}}). \end{aligned} \tag{8}$$

The energy spent on the control-plane routines is

$$\Delta W = T_{\text{DP}}(P_{\text{DP}} + P_{\text{idle}}) + T_{\text{PM}}(P_{\text{PM}} + P_{\text{idle}}) + T_{\text{BO}}(P_{\text{BO}} + P_{\text{idle}}), \tag{9}$$

and the fraction of control-plane energy is

$$\frac{\Delta W}{W} = \frac{T_{\text{DP}}(P_{\text{DP}} + P_{\text{idle}}) + T_{\text{PM}}(P_{\text{PM}} + P_{\text{idle}}) + T_{\text{BO}}(P_{\text{BO}} + P_{\text{idle}})}{(T - T_{\text{BO}} - T_{\text{DP}} - T_{\text{PM}})(P_{\text{ag}} + P_{\text{idle}}) + T_{\text{DP}}(P_{\text{DP}} + P_{\text{idle}}) + T_{\text{PM}}(P_{\text{PM}} + P_{\text{idle}}) + T_{\text{BO}}(P_{\text{BO}} + P_{\text{idle}})}. \tag{10}$$

Since BO/DP/measurement are lightweight compared to draft decoding, their incremental powers are typically smaller, i.e., $P_{\text{BO}} < P_{\text{ag}}$, $P_{\text{DP}} < P_{\text{ag}}$, and $P_{\text{PM}} < P_{\text{ag}}$. Equivalently,

$$P_{\text{BO}} + P_{\text{idle}} \leq P_{\text{ag}} + P_{\text{idle}}, \quad P_{\text{DP}} + P_{\text{idle}} \leq P_{\text{ag}} + P_{\text{idle}}, \quad P_{\text{PM}} + P_{\text{idle}} \leq P_{\text{ag}} + P_{\text{idle}}. \tag{11}$$

Let $S \triangleq T_{\text{BO}} + T_{\text{DP}} + T_{\text{PM}}$. From (10), we can rewrite

$$\frac{\Delta W}{W} = \frac{\Delta W}{(T - S)(P_{\text{ag}} + P_{\text{idle}}) + \Delta W} = \frac{1}{1 + \frac{(T-S)(P_{\text{ag}} + P_{\text{idle}})}{\Delta W}}. \tag{12}$$

Moreover, by (11), the control-plane energy satisfies

$$\begin{aligned} \Delta W &= T_{\text{DP}}(P_{\text{DP}} + P_{\text{idle}}) + T_{\text{PM}}(P_{\text{PM}} + P_{\text{idle}}) + T_{\text{BO}}(P_{\text{BO}} + P_{\text{idle}}) \\ &\leq (T_{\text{DP}} + T_{\text{PM}} + T_{\text{BO}})(P_{\text{ag}} + P_{\text{idle}}) = S(P_{\text{ag}} + P_{\text{idle}}). \end{aligned} \tag{13}$$

Plugging (13) into (12) yields

$$\frac{(T - S)(P_{\text{ag}} + P_{\text{idle}})}{\Delta W} \geq \frac{(T - S)(P_{\text{ag}} + P_{\text{idle}})}{S(P_{\text{ag}} + P_{\text{idle}})} = \frac{T - S}{S}, \tag{14}$$

and thus

$$\frac{\Delta W}{W} \leq \frac{1}{1 + \frac{T-S}{S}} = \frac{S}{T} = \frac{T_{\text{BO}} + T_{\text{DP}} + T_{\text{PM}}}{T}. \tag{15}$$

**Instantiating the bound with our measurements.** To connect the bound to our experimental setup, consider the first $R = 1000$ speculative rounds in Scenario 1. Let $T_{\text{tot}}$ be the total wall-clock time of these $R$ rounds, and let $T_{\text{BO,tot}}, T_{\text{DP,tot}}$, and $T_{\text{PM,tot}}$ be the total runtimes spent on BO, DP, and parameter measurement within the same period, respectively. Over $R = 1000$ speculative rounds, the control-plane energy fraction satisfies

$$\frac{\Delta W_{\text{tot}}}{W_{\text{tot}}} = \frac{R\,\Delta W}{R\,W} \leq \frac{R\,S}{R\,T} = \frac{T_{\text{BO,tot}} + T_{\text{DP,tot}} + T_{\text{PM,tot}}}{T_{\text{tot}}} = \frac{T_{\text{BO,tot}}}{T_{\text{tot}}} + \frac{T_{\text{DP,tot}}}{T_{\text{tot}}} + \frac{T_{\text{PM,tot}}}{T_{\text{tot}}}. \tag{16}$$

where $\Delta W_{\text{tot}}$ and $W_{\text{tot}}$ denote the total control-plane energy and total energy over the $R$ rounds, respectively. Using the profiling results in Table 5, we have

$$\frac{T_{\text{BO,tot}}}{T_{\text{tot}}} \leq 1.1\%, \quad \frac{T_{\text{DP,tot}}}{T_{\text{tot}}} \leq 0.013\%, \quad \frac{T_{\text{PM,tot}}}{T_{\text{tot}}} \leq 0.4\%.$$

Therefore, the control-plane energy fraction is bounded by

$$\frac{\Delta W_{\text{tot}}}{W_{\text{tot}}} \leq 1.1\% + 0.013\% + 0.4\% = 1.513\%.$$

This indicates that the additional energy consumption on the edge due to PipeSD's control-plane logic is at most 1.513% of the total edge energy consumption in our experiments, which is negligible.

*Table A.3.* Average TPT (ms) under one-to-many cloud-edge deployment with fluctuating network conditions.

| Clients | Vanilla | PipeSD | Reduction (%) |
|---|---|---|---|
| 2 | 83.51 | 52.51 | 37.1 |
| 4 | 37.88 | 29.21 | 22.9 |
| 8 | 20.96 | 14.13 | 32.6 |

## I. Extension to Multi-Edge Deployment

In practical cloud-edge collaborative inference scenarios, there may be multiple edge devices communicating with a single cloud server. Although PipeSD is described for a single edge device, it naturally generalizes to the multi-edge setting. Specifically, each edge device independently runs an instance of PipeSD, managing its own draft model, transmission controller, communication interface, environment monitor, and parameter updater. The cloud server maintains a single communication API and target model, handling requests from all edge devices.

To further validate the effectiveness of PipeSD in this setting, we additionally evaluate it under a one-to-many deployment with one cloud server and multiple edge clients. We consider a fluctuating-network setting in Scenario 4, where the communication bandwidth changes over time during inference. All clients are launched simultaneously and issue requests asynchronously under a full-load setting, i.e., each client starts a new inference task immediately after finishing the previous one.

Table A.3 shows that PipeSD consistently outperforms the vanilla cloud-edge speculative decoding baseline in the one-to-many setting. Specifically, PipeSD reduces TPT by 37.1%, 22.9%, and 32.6% for 2, 4, and 8 clients, respectively. These results indicate that PipeSD remains effective under concurrent multi-edge requests and fluctuating communication conditions. The gain mainly comes from two aspects: (i) the token-batch pipeline scheduling mechanism still helps overlap draft-token generation and transmission under dynamic network conditions, and (ii) the dual-threshold NAV triggering mechanism remains effective in determining verification triggering timing when multiple clients compete for cloud-side service resources. Overall, these results demonstrate that PipeSD can be extended from single-edge deployment to practical one-to-many cloud-edge serving scenarios.

## J. Discussion of Tree-based Speculative Decoding

Tree-based speculative decoding (Miao et al., 2024) is an advanced speculative decoding approach that generates multiple draft token sequences in parallel, forming a tree of candidate continuations. Although this design can further reduce the frequency of NAV calls, it imposes substantial computational demands on edge devices and may result in the transmission of a large number of speculative tokens. As a consequence, tree-based speculative decoding can incur significant bandwidth overhead, particularly in bandwidth-constrained cloud-edge settings. Therefore, tree-based speculative decoding is less suitable for the cloud-edge deployment scenario considered in this work; nonetheless, PipeSD's techniques such as token-batch pipeline scheduling and dual-threshold NAV triggering can be adapted to enhance tree-based speculative decoding in future work.

