# OpenReview forum: "PipeSD: An Efficient Cloud-Edge Collaborative Pipeline Inference Framework with Speculative Decoding"
_ICML.cc/2026/Conference — ICML 2026 regular_

### Official Review · Reviewer_dKqZ · 2026-03-06

**Soundness:** 3
**Presentation:** 3
**Significance:** 2
**Originality:** 2
**Overall Recommendation:** 2
**Confidence:** 5

**Summary:**

The paper examines cloud–edge collaborative inference for large language models, focusing on the inefficiencies introduced by sequential computation–communication execution and rigid verification triggering in speculative decoding frameworks. It presents a comprehensive solution that combines algorithmic design with system-level implementation.

The manuscript investigates the interaction between token-batch pipeline scheduling and dual-threshold verification triggering. By overlapping computation and communication and adaptively adjusting verification timing based on both token-level and sequence-level confidence, the proposed design mitigates performance bottlenecks in conventional cloud–edge speculative decoding systems.

**Compliance With Llm Reviewing Policy:**

Affirmed.

**Final Justification:**

My key concerns remain unresolved. The evaluation relies on partially emulated settings with strong assumptions, making it unclear whether the reported gains hold in realistic deployments.
﻿

In addition, the comparison with the cloud-only baseline may not be fully fair, and several results (e.g., similar latency across model scales) are insufficiently explained.
﻿

Overall, the current evidence is not yet sufficient to support the claimed system-level benefits.

**Key Questions For Authors:**

1. Can the authors further clarify and substantiate the motivation for the cloud–edge deployment setting? In particular, are there representative real-world scenarios where cloud–edge speculative decoding is a common or compelling deployment paradigm?
2. How does deploying the draft model on the edge compare against a cloud-only speculative decoding setup in terms of cost, resource utilization, and performance? Could the authors provide a quantitative comparison or discussion to justify this design choice?
3. The experiments focus on target models around 7B parameters. How would the proposed scheduling and triggering mechanisms scale to substantially larger models (e.g., tens of billions of parameters), where verification latency and communication–computation trade-offs may differ significantly?

**Limitations:**

Yes

**Strengths And Weaknesses:**

Strengths:
- The paper is clearly written and provides a precise formulation of the problem it addresses.
- The paper formalizes cloud–edge speculative decoding and frames token-batch scheduling as an explicit optimization problem.

Weaknesses:

While the technical design is carefully implemented and empirically validated, several aspects limit the broader impact and practical justification of the work:
- The motivation for the cloud–edge deployment setting would benefit from further justification, as cloud–edge speculative decoding does not yet appear to be a broadly adopted paradigm, and representative real-world use cases are not clearly discussed.
- The paper does not analyze the cost of deploying the draft model on the edge or compare against a cloud-only setup to justify this design choice.
- The experiments are limited to target models around 7B parameters, leaving the scalability of the proposed mechanisms to substantially larger models unclear.

---

> ### Author Rebuttal · Authors · 2026-03-29
>
> We thank Reviewer dKqZ for the constructive feedback. We address each point below and will incorporate the clarifications and results into the revision if accepted.
>
> **[W1,Q1]**
>
> We agree that cloud-edge speculative decoding is still an emerging paradigm, rather than a broadly standardized deployment practice. At the same time, it is not a purely hypothetical setting: there is already a small but growing line of work on edge-cloud collaborative LLM inference, including Hybrid SLM and LLM for Edge-Cloud Collaborative Inference[1] and, more directly, SpecEdge[2]. In particular, **SpecEdge**[(https://openreview.net/forum?id=4QVLKwgg3S)] was selected as a **NeurIPS 2025 Spotlight**, which in our view is strong evidence that the community recognizes this setting as timely and important. We will revise the paper to make this positioning clearer: our claim is not that cloud-edge speculative decoding is already mainstream, but that it is an emerging and increasingly validated systems direction.
>
> The motivation for applying speculative decoding to edge-cloud collaboration is exactly the one emphasized by SpecEdge: consumer-grade GPUs at the network edge create a compelling opportunity to reduce LLM serving cost by **offloading computation onto low-cost and widely available edge resources**. This motivation is especially natural for speculative decoding, because its two-model structure already matches the edge/cloud resource hierarchy: a small draft model can run on weaker edge devices, while a large target model remains in the cloud. The edge therefore contributes inexpensive draft tokens, and the cloud retains the stronger model for exact verification. SpecEdge provides direct empirical support for this view, reporting higher cost efficiency and server throughput compare with cloud-only deployment. Beyond this main economic motivation, the same deployment pattern also naturally supports **selective uploading for privacy-sensitive requests** and **graceful fallback under weak connectivity**, since the edge-side small model can continue local inference even when the cloud is slow or unavailable.
>
> Representative scenarios are concrete. A particularly important one is **future cloud-edge compute collaboration**: as LLM serving demand grows, widely distributed and underutilized edge GPUs across homes, enterprises, and access networks can serve as draft-side compute, rather than relying solely on centralized cloud accelerators. This setting is also relevant to **personal mobile assistants**, **health or wearable AI assistants**, and **vehicular/mobile edge environments** with unstable networks. In these cases, cloud-edge speculative decoding provides a natural operating point: the edge performs lightweight drafting near the user or available compute resources, while the cloud is invoked only when stronger verification is needed.
>
> [1] Hybrid SLM and LLM for Edge-Cloud Collaborative Inference, EdgeFM '24
>
> [2] SpecEdge, NeurIPS'25
>
>
> **[W2,Q2]**
>
> Deploying the draft model on the edge is more cost-effective than a pure-cloud setup, as consumer-grade GPUs offer comparable compute at significantly lower cost [1]. Moreover, edge compute is often already available but underutilized.
>
> We conducted two experimental setups: (1) a cloud-edge collaborative deployment where the cloud server coordinates with 16 edge clients, and (2) a cloud-only deployment consisting of one target model and 16 draft models. The experimental results are as follows.
>
>   | Method | Throughput (tok/s) | Energy / 1K tokens (J) |
>   |---|---:|---:|
>   | Cloud-only | 119.99 | 1091.61 |
>   | Cloud-edge | 168.10 | 794.19 |
>
>   Compared with pure-cloud speculative decoding, the cloud-edge variant improves throughput by 40.1% while reducing energy consumption by 27.3%. These results support our claim that, cloud-edge speculative decoding can simultaneously provide higher serving throughput and lower energy cost than a cloud-only deployment. This observation is also consistent with the findings reported in SpecEdge, which likewise suggest the potential efficiency advantages of leveraging edge-side drafting rather than relying entirely on centralized cloud execution.
>
> **[W3,Q3]**
>
> To test scalability, we use a 33B cloud model and repeat the evaluation. The results are very similar to those with 6.7B:
>
> | Algorithm | 6.7B TPT (ms) | 33B TPT (ms) |
> | --- | ---: | ---: |
> | vanilla | 212.530 | 213.313 |
> | hsl | 156.023 | 156.085 |
> | edgeLLM | 157.372 | 158.079 |
> | pipesd | 131.869 | 132.560 |
>
> Due to the limited rebuttal time, we were unable to deploy even larger models in this round, but we agree this is an important direction and will investigate it in future work.
>
> From a theoretical perspective, scaling up the cloud model has two opposing effects: slower cloud verification reduces the relative gain from pipelining, while making accurate NAV triggering more valuable by increasing the cost of unnecessary verification.

---

> > ### Author Rebuttal · Reviewer_dKqZ · 2026-04-02
> >
> > We have a few questions regarding the experimental setup and reported results:
> >
> > (1) Could the authors provide more details on how requests arrive in the experiments, particularly in the setting where the cloud server coordinates with 16 edge clients? For example, are requests generated synchronously or asynchronously, and what workload or arrival distribution is used?
> >
> > Again: In the original experimental setup that the cloud device appears to serve only a single client, a well-organized cloud-only setup could provide strong performance. It is unclear whether a cloud-edge approach would still provide significant advantages.
> >
> > (2) It is unclear what specific 6.7B and 33B models are used in the new evaluation. In addition, the reported TPT (ms) results for these two model sizes are nearly identical across all methods, which is somewhat unexpected. Could the authors clarify the underlying reason for this observation (e.g., whether the system is bottlenecked by other factors)?
> >
> > (3) To improve reproducibility, it would be helpful to provide more detailed experimental configurations, including hardware setup, workload characteristics, and whether results are averaged over multiple runs.
> >
> > (4) Regarding the energy evaluation, while edge devices are often considered energy-efficient, CPUs are generally less energy-efficient than GPUs for LLM inference. Could the authors clarify how energy consumption is measured and whether this factor is properly accounted for?
> >
> > Overall, additional clarification on these aspects would help strengthen confidence in the reported results.

---

> > > ### Author Response · Authors · 2026-04-03
> > >
> > > We thank the reviewer for the thoughtful questions and constructive suggestions. We sincerely appreciate the time and effort devoted to evaluating our work. All suggestions that can be incorporated will be reflected in the revised manuscript if accepted.
> > >
> > > **[Q1]**
> > >
> > > Each client generates requests asynchronously. HumanEval samples are partitioned across edge clients. All clients start at roughly the same time, and thereafter each client submits a new inference request only according to its own interaction with the cloud, without coordinating with others. As a result, after startup, the server observes interleaved asynchronous requests from 16 clients.
> > >
> > > We use a full-load setting: once a client finishes its current task, it immediately starts the next assigned one. Thus, the server remains continuously loaded during the experiment.
> > >
> > > **[Q1:again]**
> > >
> > > We agree that in the single-client setting, a well-optimized cloud-only system can be strong.  However, there are practical cases where a cloud-only solution is not reliable enough, while a cloud-edge system can still keep serving requests. In particular, under privacy-sensitive settings or unstable networks, cloud-only execution may become unavailable because it relies on continuous data transmission and stable connectivity. In contrast, PipeSD can fall back to local execution with the edge-side draft model.
> > >
> > > Beyond this, the cloud-edge setting has two system-level advantages.
> > > First, cloud-edge collaboration reduces cloud-side load, energy, and cost by offloading draft generation to otherwise idle edge resources, while reserving the cloud for verification.
> > > Second, it scales better in the multi-client setting: multiple edge clients can draft in parallel while sharing cloud verification resources.
> > >
> > > **[Q2]**
> > >
> > > The 6.7B model is DeepSeek-Coder-6.7B-Instruct and the 33B model is DeepSeek-Coder-33B-Instruct.
> > >
> > > The main reason of the observation is that, in our setup, the cloud server uses an NVIDIA A800 GPU, whose compute capacity is sufficient for both models under our tested workload.
> > > In addition, the acceptance rates under the 6.7B and 33B target models are very close. Thus, end-to-end token latency is influenced more by system-level factors, such as the pipeline upload policy and NAV triggering, than by target-model size itself. As a result, the runtime gap is much smaller than one might expect.
> > >
> > > **[Q3]**
> > >
> > > For **hardware setup**, due to rebuttal-time and resource constraints, we did not run a fully physical one-to-16-client deployment with 16 independent edge devices. Instead, we used a controlled emulation on four A800 servers: one cloud server and three servers emulating edge clients with 5, 5, and 6 client instances, respectively.
> > >
> > > To emulate realistic edge computation, we first measured the actual edge generation speed and obtained an average of 36 ms/token, then injected the corresponding delay into A800-based inference. We also injected communication delay corresponding to the 2.5 MB/s bandwidth setting.
> > >
> > > For **workload characteristics**, we use HumanEval and evenly partition samples across clients. All clients start at approximately the same time, generate requests asynchronously, and immediately start the next assigned task after finishing the current one.
> > >
> > > Regarding **repeated measurements**, all reported results are averaged over 1000 speculative rounds, as stated in the paper.
> > >
> > > **[Q4]**
> > >
> > > We agree that CPUs are generally less energy-efficient than GPUs for LLM inference. In this work, however, we measured only cloud-side energy consumption because cloud GPU energy can be measured relatively reliably via NVIDIA NVML [1], whereas edge-side CPU energy is much harder to measure accurately due to background OS activities, shared processes, and other hardware components [2]. Therefore, we did not measure edge-side CPU energy consumption.
> > >
> > > In addition, under cloud-edge collaboration, edge-side compute is often local idle resource with much lower effective cost than cloud GPU computation [3]. Therefore, we primarily focus on cloud-side energy, which is more directly related to actual monetary cost.
> > >
> > > This scope is also consistent with prior work [4,5], which similarly focuses on GPU energy overhead without considering CPU energy consumption. Following this convention, we report cloud-side ECS as the primary energy metric to capture how much PipeSD reduces the cloud energy burden.
> > >
> > > We also measured the energy consumption of the cloud-only baseline in the single-client setting, consistent with Scenario 1 in the paper. PipeSD reduces cloud-side energy consumption by **46%** compared with cloud-only deployment.
> > >
> > > **References**
> > >
> > > [1] NVIDIA. *NVIDIA Management Library (NVML)*.
> > > [2] Kasioulis et al. *Power Estimation Models for Edge Computing Devices*. Euro-Par Workshops.
> > > [3] Park et al. *SpecEdge*. NeurIPS.
> > > [4] Luccioni et al. *Estimating the Carbon Footprint of BLOOM*. JMLR.
> > > [5] Gao et al. *PipeSFL*. IEEE TMC.

---

### Official Review · Reviewer_xGfG · 2026-03-08

**Soundness:** 3
**Presentation:** 3
**Significance:** 2
**Originality:** 2
**Overall Recommendation:** 4
**Confidence:** 4

**Summary:**

This paper presents PipeSD, a cloud-edge collaborative inference framework for large language models (LLMs) that integrates speculative decoding. The authors identify two key performance bottlenecks in existing frameworks: (1) **sequential computation-communication execution**, where draft token generation and transmission are serialized, resulting in low utilization of both bandwidth and computing resources;  (2) **inflexible non-autoregressive verification (NAV) triggering**,  where fixed draft lengths or single-threshold confidence lead to either premature verification or excessive speculation.

To address these issues, PipeSD introduces two core mechanisms: (1) **a token-batch pipeline scheduling mechanism** which overlaps draft token generation and communication by formulating and solving a dynamic programming optimization problem for optimal token batching; (2) **a dual-threshold NAV triggering mechanism** that jointly considers sequence-level and token-level confidence, with a lightweight Bayesian optimization tuner for automatic threshold adaptation

The framework is evaluated on a real-world cloud-edge testbed with two draft-target model pairs across four experimental scenarios, achieving 1.16×–2.16× speedup and reducing cloud-side energy consumption by 14.3%–25.3% over state-of-the-art baselines.

**Compliance With Llm Reviewing Policy:**

Affirmed.

**Key Questions For Authors:**

**1.** The per-token generation time γ is assumed approximately constant within a 200-token sliding window (Figure 6b). What is the empirical variation of γ on the IoT-class device (Scenario 3) across the full sequence lengths observed in the experiments? If γ varies significantly, how frequently is the DP scheduler re-triggered, and does this introduce meaningful overhead?

**2.** In Scenario 4, how frequently is the BO autotuner re-triggered due to bandwidth changes? What is the average TPT during each cold-start convergence window relative to steady-state TPT? Without this data, the robustness claim in Section 5.2.1 is not fully substantiated.

**3.** Could SpecEdge be evaluated in a single-edge configuration under the same testbed? If not, please provide a more rigorous technical justification for its exclusion rather than the current one-sentence claim of orthogonality.

**4.** For target models at 13B or 70B parameters, NAV latency increases substantially relative to communication latency. Is there any theoretical or empirical analysis indicating that pipeline scheduling remains beneficial at these scales?

**Limitations:**

The paper's discussion of limitations is insufficient. Specifically, the main text should address: (1) the boundary conditions of the γ-constant assumption and its applicability under resource-constrained or long-context settings; (2) the single-client constraint as a substantive practical deployment limitation; (3) the restricted model scale and its implications for the generalizability of the results.

The paper lacks a dedicated limitations section. Adding a limitations paragraph in the main text would help readers more accurately assess the scope and applicability of this work.

**Strengths And Weaknesses:**

**Strengths：**

- The pipeline scheduling problem is rigorously formulated as a mathematical optimization problem, and the optimality of the DP algorithm is formally proven (Theorem 4.1). This level of theoretical rigor distinguishes PipeSD from purely empirical systems work.
- Identifying communication startup overhead (α) as the key factor behind suboptimal token-level pipelining is a sharp insight, directly validated by empirical measurement (Figure 6a).
- The dual-threshold NAV triggering mechanism is a principled response to clearly articulated limitations of prior schemes. The analysis of HSL's and EdgeLLM's respective failure modes is accurate, and the dual-threshold design directly addresses both gaps.
- Evaluation on a real metropolitan network rather than simulation substantially strengthens the practical credibility of the results. The inclusion of cloud-side energy consumption (ECS) as a metric reflects meaningful attention to deployment cost beyond raw latency.

**Weaknesses：**

- The individual techniques employed — pipeline scheduling, dynamic programming, and Bayesian optimization are all well-established. The contribution is essentially a domain-specific integration of existing methods rather than novel methodology, representing relatively modest originality for ICML.
- The scope of evaluation is limited. The largest target model tested is only 6.7B parameters. For larger models e.g., 13B, 70B, NAV latency increases substantially and the relative benefit of pipeline scheduling may diminish. The paper makes no attempt to analyze behavior at larger scales.
- The assumption that per-token generation time γ remains approximately constant is central to the DP model's correctness, but the paper does not adequately discuss whether this holds under diverse conditions, particularly on the IoT-class device in Scenario 3 or for longer sequences.
- The BO auto tuner requires approximately 16 samples to converge after each retrigger. In Scenario 4 with 20-second bandwidth change intervals, the frequency of BO re-triggering and resulting TPT degradation during cold-start are never quantified, undermining the robustness.
- The exclusion of HAT and SpecEdge from the experimental comparison is justified only by a brief claim of "orthogonality," without sufficient technical elaboration. It is unclear whether a single-edge configuration of SpecEdge could have been evaluated under the same testbed.
- The single-client assumption is a substantive practical limitation but receives only brief acknowledgment in Appendix H. The paper does not analyze how concurrent NAV requests would affect the dual-threshold triggering logic.

---

> ### Author Rebuttal · Authors · 2026-03-29
>
> We thank Reviewer xGfG for the constructive feedback. We address each point below and will incorporate the clarifications and results into the revision if accepted.
>
> **[W1]**
>
> We apologize that our current presentation may have made PipeSD appear to be a simple integration of existing techniques. We agree that pipeline scheduling, DP and BO are well-established tools. However, they are not the main novelty of our work.
>
> The main novelty of PipeSD is two cloud-edge speculative decoding mechanisms tailored to this setting:
> (1) a token-batch pipeline scheduling formulation that explicitly optimizes the overlap between edge drafting and token transmission.
> Unlike traditional pipeline designs, our pipeline is built upon a rigorous mathematical formulation of the cloud-edge interaction, from which we derive the theoretically optimal solution, just as you also pointed out in Strengths 1; and
> (2) a dual-threshold NAV triggering mechanism that jointly uses token-level and sequence-level confidence.
>
> DP and BO are only tools used to instantiate these mechanisms. Our actual contribution is to identify these two bottlenecks in cloud-edge speculative decoding, formulate them precisely, and design effective solutions. We will revise the Introduction to more clearly highlight our contributions.
>
>
> **[W2,Q4]**
>
> To test scalability, we use a 33B cloud model and repeat the evaluation. The results are very similar to those with 6.7B:
>
> | Algorithm | 6.7B TPT (ms) | 33B TPT (ms) |
> | --- | ---: | ---: |
> | vanilla | 212.530 | 213.313 |
> | hsl | 156.023 | 156.085 |
> | edgeLLM | 157.372 | 158.079 |
> | pipesd | 131.869 | 132.560 |
>
> Due to the limited rebuttal time, we were unable to deploy even larger models in this round, but we agree this is an important direction and will investigate it in future work.
>
> From a theoretical perspective, scaling up the cloud model has two opposite effects. On the one hand, the benefit of pipeline overlap itself does not increase with cloud model size; moreover, a slower cloud-side verification stage increases the end-to-end latency, which can reduce the relative gain from pipelining. On the other hand, as cloud verification becomes more expensive, accurate NAV triggering becomes more important. In this regime, our dual-threshold triggering mechanism can better avoid unnecessary verification and thus provides larger value.
>
>
> **[W3,Q1]**
>
> We agree that on IoT-class devices, the empirical fluctuation of $\gamma$ is larger than on more capable edge devices. However, it can still be treated as approximately constant within a smaller sliding window, e.g., around 60 tokens. In this case, a smaller window can be used to track its update more closely, which may lead to more frequent DP re-triggering. However, the overhead is negligible: forcing a DP execution after every speculative round increases runtime by only 0.0163\%, with an average DP cost of 27.1 $\mu$s per call. We will clarify this scope explicitly in the revision.
>
> **[W4,Q2]**
>
> In Scenario 4, the bandwidth changes every 20 s, the BO re-trigger frequency is 0.05 Hz.
>
> BO autotuning is executed asynchronously, so re-triggering itself does not directly add TPT overhead. The practically relevant effect is the temporary performance drop during the BO cold-start phase. The cold-start phase takes 13.98 s on average; throughput is 6.1355 tokens/s during cold-start versus 6.8483 tokens/s after convergence, corresponding to a 10.41% degradation. We will add these quantitative results in the revision.
>
> **[W5,Q3]**
>
> We agree that our current explanation for excluding HAT and SpecEdge is too brief and will clarify it in the revision.
>
> HAT targets a different collaborative inference regime: it is based on hidden-state exchange and prompt-chunk scheduling, whereas PipeSD studies token-based cloud-edge speculative decoding with drafting and token upload timing. Thus, the communication object and optimization problem are different.
>
> For SpecEdge, its full design includes both proactive edge drafting and a server-side multi-user pipeline scheduler. In our single-client setting, the latter component is inactive, so the comparison mainly reduces to continue-drafting-during-wait behavior, which is already covered by our EdgeLLM baseline. We will make this distinction explicit in the revision.
>
> **[W6]**
>
> Concurrent NAV requests do not change the dual-threshold triggering logic itself, because the trigger is client-side and depends only on the local token-level confidence and sequence-level confidence of that client. In this sense, concurrency does not change the semantics of the NAV rule.
>
> In our one-to-many experiments under dynamic and heterogeneous settings, compared with baseline, PipeSD reduces TPT by 37.1%, 22.9%, and 32.6% at 2, 4, and 8 clients, respectively. Please refer to our response to Reviewer wx9q (Weakness 2) for further details on the multi-edge evaluation.

---

> > ### Author Rebuttal · Reviewer_xGfG · 2026-04-01
> >
> > Thank you for your rebuttal. I have no more questions and I'd like to maintain my score.

---

> > > ### Author Response · Authors · 2026-04-02
> > >
> > > Thank you for your thoughtful follow-up. We sincerely appreciate your time and are glad that our rebuttal has addressed your concerns.

---

### Official Review · Reviewer_wx9q · 2026-03-12

**Soundness:** 3
**Presentation:** 3
**Significance:** 3
**Originality:** 3
**Overall Recommendation:** 5
**Confidence:** 3

**Summary:**

The paper proposes PipeSD - a cloud-edge collaborative LLM inference framework that uses speculative decoding. In this setting the small draft model is located on the edge device for fast autoregressive inference, while the larger model is deployed on the cloud for non-autoregressive verification (NAV). Existing frameworks suffer mainly from two problems: 1) inefficient resource utilization due to the sequential token generation, communication and verification pipeline; 2) suboptimal NAV triggering. To address these shortcomings, PipeSD introduces two components: 1) token-batch scheduling mechanism designed to overlap token generation and communication; 2) dual-threshold verification triggering mechanism that takes into consideration the probabilities of each token individually as well as their product and estimates the thresholds using Bayesian optimization. Experiments show that PipeSD outperforms several SOTA baselines in terms of latency and energy consumption reduction.

**Compliance With Llm Reviewing Policy:**

Affirmed.

**Final Justification:**

My concers were addressed during the rebuttal, I believe this is a solid contribution.

**Key Questions For Authors:**

1. All the provided metrics are averaged over 1000 tokens. Given that PipeSD uses dynamic scheduling and triggering, latency might fluctuate a lot. Could the authors provide standard deviation or per-token values to confirm that the performance is stable?

2. Appendix D discusses how thresholds $\delta_1$, $\delta_2$ and $\delta_3$ are used to trigger the updates. I'd assume the method can be quite sensitive to the choice of the thresholds. Could the authors provide empirical evidence or guidelines on how to select these thresholds, and an analysis of how sensitive the performance is to their choice?

**Limitations:**

Yes

**Strengths And Weaknesses:**

## Strengths

1. The proposed method is quite simple which I see as a benefit. All the introduced components can be integrated easily, add minimal overhead and are well-motivated.
2. The framework is designed to be adaptive to the dynamically changing conditions, and the paper provides the details of how the dynamic coefficients are estimated and when they need to be updated.
3. The set of experiments is extensive. The method is tested in several real-world like scenarios, baselines are reasonable, ablations for all the introduced components are provided. Furthermore, the authors provided the code and all the necessary experimental details for reproducibility.
4. The paper is very clear. It gives enough background for the reader, explains the motivation well and all the claims are supported by experiments.


## Weaknesses

1. There is one part in the introduction that might be confusing. When authors discuss that existing frameworks have two limitations, it gives the impression that these topics were not yet  investigated at all. I'd appreciate if the authors could briefly clarify that there are existing attempts as solving them. For example, list a few papers that are later discussed in Section 2.4. Further, some of the papers that tackle inflexible NAV triggering are missing from the background review [1,2,3]. It is important to note, that even if NAV triggering is not fully explored in cloud-edge setting, all the literature from the broader speculative decoding area is still relevant because the triggering decision concerns the draft side of the framework only. Regarding the proposed pipeline scheduling, it is important to emphasize that your method focuses on optimizing the communication part specifically, because overlapping generation and verification/feedback waiting has been explored [4,5] and is orthogonal to your contribution.

2. The paper doesn't address the multi-edge deployment which is the more common scenario in real world. Although the authors suggest that their method can be directly extended to multi-edge deployment in Appendix H, it is not clear how a dynamic heterogeneous setting will affect all the estimations when the server is under high load.

[1] Huang, Kaixuan, Xudong Guo, and Mengdi Wang. "SpecDec++: Boosting Speculative Decoding via Adaptive Candidate Lengths." Second Conference on Language Modeling.

[2] Li, Xiangchen, et al. "Sled: A speculative llm decoding framework for efficient edge serving." Proceedings of the Tenth ACM/IEEE Symposium on Edge Computing. 2025.

[3] Zhang, Ziyin, et al. "Draft model knows when to stop: Self-verification speculative decoding for long-form generation." Proceedings of the 2025 Conference on Empirical Methods in Natural Language Processing. 2025.

[4] Park, Jinwoo, Seunggeun Cho, and Dongsu Han. "Specedge: Scalable edge-assisted serving framework for interactive llms." arXiv preprint arXiv:2505.17052 (2025).

[5] Venkatesha, Yeshwanth, Souvik Kundu, and Priyadarshini Panda. "Fast and cost-effective speculative edge-cloud decoding with early exits." arXiv preprint arXiv:2505.21594 (2025).

---

> ### Author Rebuttal · Authors · 2026-03-28
>
> We thank the reviewer wx9q for the constructive comments. We will revise the paper to better position our contribution, strengthen the discussion of multi-edge deployment, and add additional results on runtime stability and threshold sensitivity.
>
> **[W1]**
>
> We acknowledge that our original manuscript did not clearly distinguish our pipeline design from prior pipeline approaches, and lacked sufficient discussion of related work on NAV triggering.
>
> Regarding pipeline scheduling, we agree that prior work, such as SpecEdge, has already explored pipelined execution in speculative decoding. However, the focus of our pipeline design is different. In the revision, we will clarify that our contribution is not the general idea of overlapping generation with verification, but rather a communication-aware scheduling design tailored to cloud-edge speculative decoding. We will revise the Introduction to better highlight this distinction and make the unique focus of PipeSD clearer.
>
> Regarding NAV triggering, we acknowledge that the current Introduction mainly frames this issue within the cloud-edge collaborative setting and does not sufficiently connect it to the broader speculative decoding literature. We agree that this broader line of work is relevant, since the triggering decision is fundamentally associated with the draft side itself. In the revision, we will expand the related discussion in the Introduction and add representative prior studies, including the references mentioned by the reviewer.
>
>
> **[W2]**
>
> We agree that multi-edge deployment under dynamic and heterogeneous load is an important practical setting. We conducted an additional experiment in a fluctuating network environment with multiple heterogeneous clients. We report TPT (ms):
>
> | Clients | vanilla | PipeSD |
> | --- | ---: | ---: |
> | 2 | 83.51 | 52.51 |
> | 4 | 37.88 | 29.21 |
> | 8 | 20.96 | 14.13 |
>
> These results show that PipeSD consistently outperforms vanilla across all client counts under dynamic network conditions. In particular, PipeSD reduces TPT by `37.1%`, `22.9%`, and `32.6%` at 2, 4, and 8 clients, respectively. This indicates that PipeSD continues to scale with increasing client concurrency and preserves its advantage over the vanilla baseline even in a more realistic multi-client setting. We will add this result to the revision.
>
> Moreover, the key estimation quantities in PipeSD are primarily determined by hardware characteristics and task difficulty, and are not directly affected by system-level load, heterogeneity, or temporal dynamics. The strong performance of PipeSD in multi-edge setting further suggests that our estimations remain accurate even under dynamic, heterogeneous and multi-edge conditions.
>
>
> **[Q1]**
>
> We agree that reporting only the 1000-token average is insufficient to fully capture runtime stability. Therefore, we additionally report both the mean TPT and its standard deviation to provide a more comprehensive assessment.
>
> | Algorithm | Mean (ms) | Std. (ms) |
> | --- | ---: | ---: |
> | PipeSD | 131.771 | 0.624 |
> | HSL | 156.071 | 1.563 |
> | EdgeLLM | 157.491 | 1.772 |
>
> PipeSD is not only faster on average, but also more stable, with substantially lower standard deviation than HSL and EdgeLLM. We will add these statistics to the revision and clarify that the reported averages reflect stable runtime behavior rather than a few favorable outliers.
>
> **[Q2]**
>
> We agree that the thresholds in Appendix D should be better explained.
>
> PipeSD’s update rule can be viewed as an event-triggered mechanism under noisy observations. Updates are triggered when relative changes exceed $\delta_1,\delta_2,\delta_3$, balancing responsiveness and update overhead.
>
> If thresholds are too small, noise leads to excessive false updates; if too large, true changes are detected too late. Under sub-Gaussian noise with scale $\sigma$, the false-trigger probability is bounded by
> $$
> \Pr(\text{false trigger}) \le 2\exp(-c\delta^2/\sigma^2),
> $$
> while missed triggers decrease when the true change magnitude $\Delta$ satisfies $\delta<\Delta$.
>
> Therefore, effective thresholds lie in an intermediate regime (above noise scale but below meaningful change magnitude), and PipeSD is not highly sensitive to their exact values within this range.
>
> In our experiments, we set $\delta_1=\delta_2=\delta_3=0.2$, and we found that the performance remains very similar when these thresholds vary within the range $[0.1, 0.5]$, which empirically supports this robustness.
>
> In the revision, we will add this explanation to Appendix D.

---

> > ### Author Rebuttal · Reviewer_wx9q · 2026-04-01
> >
> > I thank the authors for addressing my concerns. I'll increase the score.

---

> > > ### Author Response · Authors · 2026-04-02
> > >
> > > Thank you for your thoughtful follow-up. We sincerely appreciate your time and are glad that our rebuttal has addressed your concerns.

---

### Official Review · Reviewer_g7PG · 2026-03-13

**Soundness:** 3
**Presentation:** 3
**Significance:** 3
**Originality:** 3
**Overall Recommendation:** 4
**Confidence:** 5

**Summary:**

PipeSD is a cloud-edge speculative decoding framework that overlaps token generation and communication via a dynamic-programming-based token-batch scheduler, while using a BO-tuned dual-threshold verification policy to adapt speculative length dynamically.

**Compliance With Llm Reviewing Policy:**

Affirmed.

**Final Justification:**

In the rebuttal, the added speculative-decoding diagnostics (acceptance rate, accepted/rejected lengths, verification frequency) fully resolve my concerns about interpreting the NAV mechanism’s gains; PipeSD’s high acceptance rate and very small rejected length strongly support the claimed verification-timing benefits. On the DP scheduler, the authors addressed my main concern by implementing the suggested greedy pipelined baseline (“send all accumulated tokens when the network is idle”) and showing DP consistently achieves the best performance across a range of ((\alpha,\beta)) settings, with up to a 1.13× improvement over greedy under high startup overhead and negligible DP runtime overhead. While the DP’s advantage over strong heuristics is modest in some regimes, the new results convincingly show it provides consistent (and sometimes meaningful) gains at near-zero cost, strengthening the paper’s practical contribution.

**Key Questions For Authors:**

- In Figure 6, the measured regime $\alpha \approx 14, \beta \approx 2.5, \gamma \approx 35$ ms, appears compute-bound. In this case, why is immediate per-token transmission not already near-optimal? How should we interpret the 1.18x gain of **PipeSD + Fixed-length** if simple immediate-send could already hide most communication latency?
- Could the authors report the empirical distribution of the BO-induced draft length N, along with the realized batching decisions?

**Limitations:**

Yes

**Strengths And Weaknesses:**

**Strengths**
- The paper targets a practical and well-motivated systems problem: in cloud-edge speculative decoding, communication can easily erode the gains from local drafting, so improving compute/communication overlap is important.
- The design is clean and complementary: token-batch scheduling improves overlap between drafting and transmission, while dual-threshold NAV makes cloud verification more adaptive than fixed-length or single-threshold policies.
- The experimental section is convincing: the paper evaluates across multiple scenarios and reports strong end-to-end performance gains.

**Weaknesses**
- The practical necessity of the DP token-batching scheduler is not fully established. Since the system already measures the communication parameters ($\alpha, \beta$) and generation latency $\gamma$, a simple latency-aware heuristic may be sufficient. Without a comparison against such a greedy or threshold-based baseline, it is unclear whether the DP scheduler provides meaningful benefits beyond basic batching logic.
- The evaluation omits several standard speculative-decoding diagnostics that are needed to interpret the gains from the dual-threshold NAV mechanism, including the BO-induced draft-length distribution $N$, accepted-prefix length, rejection/rollback statistics, and verification frequency.

---

> ### Author Rebuttal · Authors · 2026-03-27
>
> We thank Reviewer g7PG for the thoughtful feedback and constructive comments. We respond to each weakness and question below. For points addressed through additional explanations or experiments in the rebuttal, we will incorporate the corresponding discussion and results into the revised paper if accepted.
>
>
> **[W1]**
>
> We apologize for not clearly explaining the necessity of the DP scheduler. To better justify its practical value, we added a controlled comparison against a simple latency-aware heuristic that switches between two basic upload policies according to the measured communication and generation latencies: it uses no-early-upload in the communication-bound regime and immediate-send otherwise. Under this comparison, PipeSD-DP achieves a TPT of 131 ms, while the heuristic baseline yields 152 ms, corresponding to a 1.16x speedup. This result shows that the DP scheduler provides a nontrivial benefit beyond a simple latency-aware heuristic.
>
>
> We further measured the overhead of the DP scheduler and found it negligible. Forcing a DP execution after every speculative round results in only a 0.0163% increase in runtime, with an average DP cost of 27.1 $\mu$s per call. Therefore, the DP scheduler is both effective and practical.
>
> **[W2]**
>
> We thank the reviewer for pointing out that the original evaluation did not report several standard speculative-decoding diagnostics that are useful for understanding the gains of the dual-threshold NAV mechanism. To address this, we added a diagnostic run that records the empirical draft length before each verification, the accepted-prefix length, the rejected length, and the verification frequency for PipeSD, vanilla, HSL, and edgeLLM.
>
> The results below are from a HumanEval diagnostic run.
>
> | Method | TPT (s) | Verification Frequency | Mean Draft Length | Mean Accepted Prefix | Mean Rejected Length | Acceptance Rate |
> | --- | ---: | ---: | ---: | ---: | ---: | ---: |
> | vanilla | 0.1883 | 0.1396 | 8.00 | 6.16 | 1.84 | 0.7704 |
> | HSL | 0.1502 | 0.2558 | 3.18 | 2.91 | 0.27 | 0.9148 |
> | edgeLLM | 0.1389 | 0.1912 | 4.74 | 4.23 | 0.51 | 0.8917 |
> | **PipeSD** | **0.1254** | **0.1733** | **4.96** | **4.77** | **0.19** | **0.9616** |
>
> These diagnostics clarify the source of PipeSD's latency gains. Vanilla is overly aggressive, with the lowest acceptance rate. HSL is too conservative, using short drafts and frequent verification, which limits speculative gains. edgeLLM is more balanced, but still has a lower accepted-prefix length and a larger rejected length than PipeSD.
>
> By contrast, PipeSD achieves the best tradeoff: it maintains a moderate draft length, while attaining the highest acceptance rate. These results suggest that the dual-threshold NAV mechanism enables more accurate verification decisions, which directly translates into lower latency.
>
>
> **[Q1]**
>
> We would like to first clarify that the scenario in Figure 6 is not actually compute-bound. In our measured setting, the relevant communication parameter is $\beta \approx 25$ ms rather than 2.5 ms, so the communication cost is of the same order as the token generation latency $\gamma \approx 35$ ms. Therefore, immediate per-token transmission is not expected to be near-optimal in this scenario.
> The experimental results provided in response to Weakness 1 indicate that the DP scheduler provides a measurable improvement over immediate per-token transmission.
>
> We would like to clarify that the 1.18x gain of PipeSD + Fixed-length is measured against the compute-first, transmit-later policy, rather than against immediate per-token transmission. Thus, this result should be interpreted as the benefit of pipelined uploading over no-early-upload.
>
>
> **[Q2]**
>
> We thank the reviewer for the suggestion. We will add the empirical distribution of the BO-induced draft length \(N\), together with the realized batching decisions, to the revision.
>
> Below are the exact realizations from our two evaluation tasks.
>
> **Task 1**
> - Draft length \(N\): `2, 2, 1, 3, 3, 1, 1, 8, 3, 6, 8, 4, 11, 21, 6, 1, 12, 1, 1, 3, 2, 1`
> - Realized batching: `1,1; 1,1; 1; 1,2; 1,2; 1; 1; 1,3,1,3; 1,2; 1,3,1,1; 1,3,1,3; 1,3; 1,1,4,1,1,3; 1,1,4,1,1,4,1,1,4,1,1,1; 1,2,3; 1; 1,2,6,1,2; 1; 1; 1,2; 1,1; 1`
> - Empirical distribution of \(N\): `1×7, 2×3, 3×4, 4×1, 6×2, 8×2, 11×1, 12×1, 21×1`
>
> **Task 2**
> - Draft length \(N\): `2, 1, 4, 7, 4, 19, 3, 3, 1, 1, 1, 6, 7, 1, 11, 1, 1, 3, 4, 3, 11, 2, 2, 1, 1, 1, 6, 7`
> - Realized batching: `1,1; 1; 1,3; 1,3,1,2; 1,3; 1,3,1,3,1,3,1,3,1,2; 1,1,1; 1,1,1; 1; 1; 1; 1,3,1,1; 1,3,1,2; 1; 1,3,1,3,1,2; 1; 1; 1,2; 1,3; 1,2; 1,3,1,3,1,2; 1,1; 1,1; 1; 1; 1; 1,3,1,1; 1,3,1,2`
> - Empirical distribution of \(N\): `1×10, 2×3, 3×4, 4×3, 6×2, 7×3, 11×2, 19×1`

---

> > ### Author Rebuttal · Reviewer_g7PG · 2026-04-02
> >
> > I thank the authors for the detailed rebuttal and additional data. The speculative-decoding diagnostics (W2) are excellent and largely resolve my concern about where the latency gains come from; PipeSD’s high acceptance rate and small rejected length strongly support the dual-threshold NAV mechanism.
> > Regarding W1, the added “latency-aware heuristic” baseline is helpful but appears too weak to isolate the marginal value of the DP optimizer. The heuristic only toggles between the two extremes—compute-first/no-early-upload (effectively a single batch of size (N)) and immediate-send (batch size 1)—and never considers intermediate pipelined batching (1<b<N). As a result, the reported 1.16× improvement may largely reflect the benefit of pipelined uploading rather than DP specifically. A stronger non-DP baseline would be a simple greedy pipelined policy: whenever the network becomes idle, transmit all currently accumulated tokens as a batch. This naturally overlaps compute and communication and adapts to ((\alpha,\beta,\gamma)) without a DP solver; in short-(N) regimes like those reported, it may already be near-optimal.

---

> > > ### Author Response · Authors · 2026-04-05
> > >
> > > We thank the reviewer for this constructive suggestion. Following the reviewer's suggestion, we implemented the mentioned baseline: a greedy pipelined policy that transmits all currently accumulated tokens whenever the network becomes idle. The results are shown below, where each entry reports the speedup of DP over the corresponding baseline under different communication startup overheads $\alpha$ and per-token transmission time $\beta$, using the same hardware setup as in the paper.
> > >
> > > | $(\alpha,\beta)$ (ms) | DP vs. Greedy | DP vs. Immediate-send | DP vs. No-early-upload |
> > > |---:|---:|---:|---:|
> > > | (20, 72)  | 1.02x | 1.09x | 1.22x |
> > > | (100, 72) | 1.04x | 1.44x | 1.10x |
> > > | (200, 72) | 1.10x | 1.76x | 1.06x |
> > > | (20, 48)  | 1.03x | 1.11x | 1.33x |
> > > | (100, 48) | 1.06x | 1.63x | 1.13x |
> > > | (200, 48) | 1.13x | 2.06x | 1.06x |
> > >
> > > These results show that DP consistently achieves the best performance across all tested settings, since it always finds the optimal batching strategy under our pipeline model. Moreover, its advantage becomes more pronounced as the communication startup overhead $\alpha$ increases. This is because larger startup overhead makes the batching strategy more important: different batching policies lead to larger performance differences, which further highlights the benefit of the optimal strategy obtained by DP.  Although the other three algorithms can be close to the optimum in some cases, our DP-based algorithm consistently remains at or near the optimum across all regimes.
> > >
> > > Since the overhead of DP itself is negligible, and it consistently provides the optimal batching strategy across all regimes, we believe it remains valuable in practice: even if the gain over heuristics is small in some cases, in other regimes it can yield clearly better performance at almost no additional cost.

---

### Decision · Program_Chairs · 2026-04-30

**Decision:**

Accept (regular)

**Comment:**

This paper introduces an effective framework for cloud-edge speculative decoding that successfully mitigates communication bottlenecks via an optimized dynamic programming scheduler. The majority of reviewers praised the clean formulation and exposition, as well as the resulting latency improvements.
I am therefore in favor of acceptance.